

# Compilation and Analysis of Thaw Settlement Test Results: Implications for Prediction Tools and Stress-Strain Characterization in Permafrost

Zakieh Mohammadi[1], Jocelyn L. Hayley[1]

[1] Department of Civil Engineering, University of Calgary, Calgary, T2N 1N4, Canada

*Correspondence to*: Zakieh Mohammadi (seyedehzakieh.mohamm@ucalgary.ca)

**Abstract.** Climate change has already significantly impacted infrastructures built on permafrost, with thaw settlement as the most frequently reported issue.  By characterizing thaw settlement tests, an improved understanding of the thaw settlement properties of permafrost sediment can be obtained. Thaw settlement tests involve thawing permafrost samples under an initial load, followed by applying additional load to the thawed sample to characterize its volume change behaviour upon thawing.

In the absence of a standardized procedure for conducting thaw settlement tests, characterizing thaw settlement properties has been done using various methods in the existing literature; however, to date, these data have not been broadly compared. This is in part because they had not previously been compiled in a single dataset. This study presents a comprehensive dataset of thaw settlement test results, digitized from published papers and reports. The data are standardized and stored in an open-source repository. Aggregating the data enabled a cross-comparison of thaw settlement properties for different soil types. This

was achieved by constructing an idealized stress-strain curve for each test and deriving thaw settlement parameters from the curves. These parameters were then used to compare the thaw settlement behaviour of fine-grained, coarse-grained, and highly organic permafrost samples. Additionally, the compiled data was used to evaluate the effectiveness of various empirical tools developed to predict thaw strain from index properties. The predicted thaw strain values were compared with the measured thaw strains to determine which tool provided the most accurate and reliable predictions for each soil type. The results suggest

that a correlation developed by Nixon and Ladanyi (1987) for estimating thaw strain based on frozen bulk density shows the smallest deviation from actual values and exhibits the least bias in its predictions. This dataset is expected to enhance the understanding of thaw settlement and improve its estimation. Used alone or in conjunction with localized data, it can contribute to developing new empirical tools for predicting thaw strain from index properties. Additionally, this dataset aids enhanced characterization of stress-strain behaviour upon thawing, facilitating future efforts in the numerical modelling of the thaw

settlement process. The dataset is accessible at https://doi.org/10.5281/zenodo.14538524, and the results presented in this paper are based on version 1.2.0 of the dataset (Mohammadi and Hayley, 2024c).



## 1 Introduction

Thaw settlement is a frequently reported issue for infrastructure built on permafrost and contributes to high maintenance costs,
reduced life cycles, and compromised serviceability of infrastructure (Fortier et al. 2011, Flynn et al. 2015, Hjort et al. 2018, 2022, Brooks 2019, Deimling et al. 2020). Thaw settlement occurs when excess water, originating from melting ground ice, is expelled under the overburden loads, resulting in ground subsidence (Andersland and Ladanyi 2003). With increasing warming trends, and projections showing a substantial decrease in near-surface permafrost extent, it is expected that thaw settlement will become an even greater threat to infrastructure (Hjort et al., 2022). Improved understanding and estimation of thaw
settlement are crucial for developing adaptive strategies, design, and management of new structures, as well as maintenance of stability and serviceability of existing infrastructure undergoing climate changes.

Thaw settlement properties at the site scale can be determined by conducting a comprehensive site investigation, obtaining an adequate number of minimally disturbed permafrost samples, and determining the thaw settlement parameters experimentally under controlled thermal and loading conditions for representative samples. This is, however, not feasible in many cases
because of remoteness, harsh weather conditions, and high costs, limiting the capacity for thaw settlement testing when planning and budgeting. In light of these challenges, empirical methods correlating thaw strain to index properties of permafrost are commonly used for predicting thaw strains. The development of new tools, and improving the existing ones relies heavily on what experimental data there is to develop correlations between thaw strain and more easily acquirable index properties of permafrost samples. Thaw settlement test results, beyond their established role in developing thaw strain
estimation tools, are also integral to characterizing the stress-strain behaviour of thawing permafrost, which is essential for accurate numerical modelling of thaw settlement. This type of data has previously been used to investigate the characteristics of stress-strain relationship in various permafrost soils (Dumais and Konrad, 2023, 2024; Mohammadi and Hayley, 2024a, b). While challenges in conducting thaw settlement tests have led to a scarcity of high-quality data, the lack of a standardized procedure for conducting the test has resulted in diverse test procedure across sources, introducing uncertainties and variability
when consolidating data for conclusive analysis. Homogenizing thaw settlement data from various sources, as well as compiling them into larger datasets, is crucial for enhancing the understanding and prediction of thaw settlement.

This paper presents a repository of thaw settlement test results sourced from the literature, that had not previously been compiled in a single dataset. This repository aims to enhance the understanding of thaw settlement behaviour across various permafrost sediments. The compilation includes thaw settlement test results, corresponding sample properties, borehole data,
and detailed particle size distribution. The thaw settlement test results and observed general stress-strain behaviour among three major groups of samples—fine-grained, coarse-grained, and organic—are discussed. The data is utilized to compare the effectiveness of existing empirical tools for estimating thaw strain and to identify the most fitting tools for various soil groups. This data provides a valuable source for inferring thaw settlement properties where site-specific data is unavailable or limited. Additionally, the compiled data is valuable for the development of new thaw strain estimation tools and the improvement of
existing ones.

## 2 Background

Thaw settlement is defined as the "settling of the ground surface, buildings, or infrastructure caused by melt of ice within the soil" (Lewkowicz et al., 2024). The magnitude of thaw settlement can vary significantly depending on soil type and ice content. The settlement continues as long as the thawing front advances and until excess water generated during thawing is expelled

and the potential pore pressure buildup, especially in relatively impermeable fine-grained soils, is dissipated.

The thaw settlement test is widely used to characterize the volume change behaviour of permafrost subjected to thawing. Figure 1 shows a typical void ratio versus vertical stress curve for a permafrost sample as it undergoes thawing and experiences additional vertical stress after thawing, obtained through this test.

Based on the theoretical concept established by Tsytovich (1960) thaw strain (measured as the percentage of volume change

from point b to c) has three main components: (1) phase change, (2) drainage of excess water (meltwater in excess of what soil can retain in the thawed state), and (3) compression of the thawed soil under the applied initial pressure during thawing ($\sigma_v$). Additional volume change may occur if an additional load is applied after thawing (points c to d).

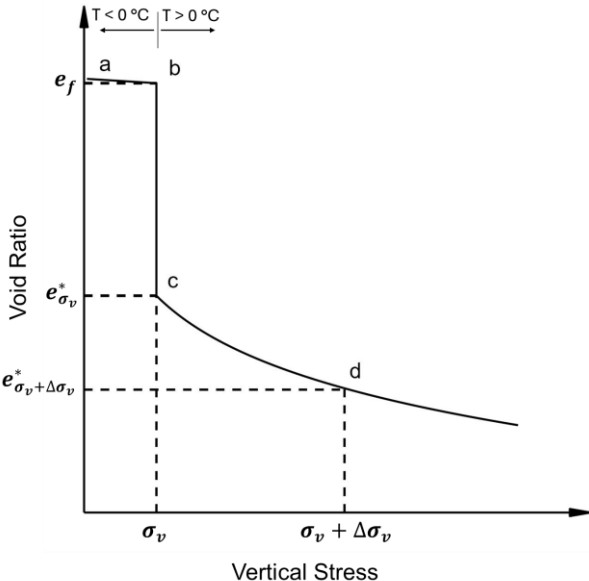

**Figure 1: Typical void ratio ($e$) versus vertical stress ($\sigma_v$) curve for a permafrost sample with a frozen void ratio of $e_f$ subjected to**
**thawing under a vertical stress of $\sigma_v$ and subjected to additional vertical stress of $\Delta\sigma_v$ after completion of thawing (adopted from** Andersland and Ladanyi 2003)**.**

The thaw settlement typically involves subjecting the sample to an initial or seating load (generally equivalent to the overburden pressure) and then increasing the temperature to induce thawing. Once thawed, and deformation is stabilized, the loading can be increased incrementally to establish stress-strain behavior under the combined effects of overburden pressure

and potential surface loading. Currently, there is no standardized procedure for conducting the test, which results in a variety

Earth System
Science
Data

of methodologies being adapted, introducing variability and uncertainty in the cross-comparison of data from different testing

programs.

For a given permafrost stratum, a thaw settlement test on representative samples measures thaw strain, which is mostly

controlled by ground ice content, soil composition, and overburden pressure. However, the magnitude of thaw settlement at

the ground surface over a specific period is influenced by stratigraphy, the thickness of the thawed layers (controlled by climate

forces and site conditions), the thaw strain of each thawed layer, and their confining conditions.

Thaw settlement test results are typically presented as either thaw strain versus vertical (effective) stress ($\varepsilon - \sigma_v$) or void ratio

versus vertical (effective) stress ($e - \sigma_v$), as shown in Figure 2. In this paper, "vertical stress" refers interchangeably to vertical

effective stress, given that sufficient time is allowed during each loading step for the dissipation of excess pore pressure and

completion of consolidation. However, in relatively permeable coarse-grained soils, the development of pore pressure is

expected to be negligible. From both representations, comparative parameters, termed thaw settlement parameters in this paper,

can be derived that enable characterization of the stress-strain behaviour and cross-comparison of the test results obtained

using various testing procedures.

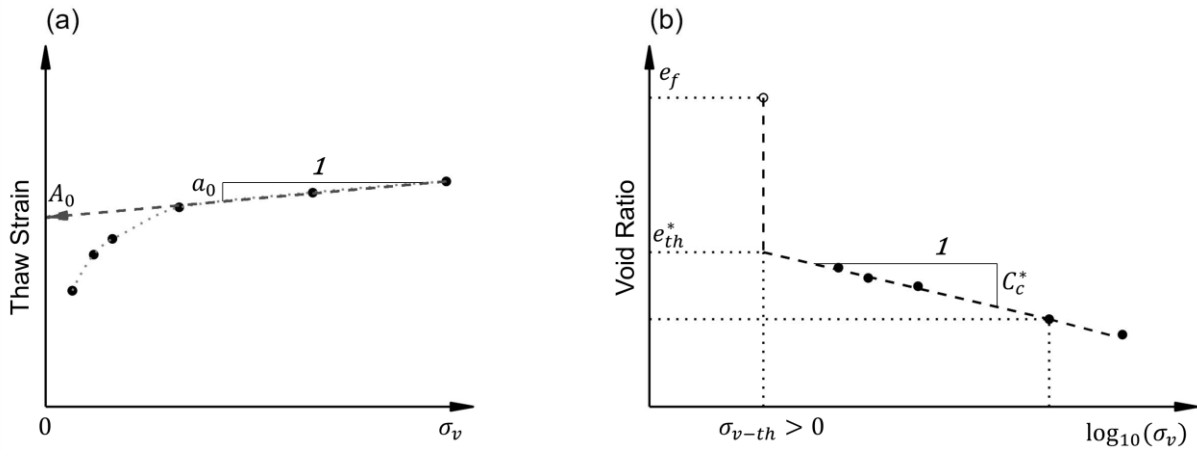

**Figure 2: Schematic of typical thaw settlement test results (a) strain versus vertical stress (b) void ratio versus logarithm of vertical stress, where "f" represents the frozen state,"*" represents the thawed state, and "th" marks the threshold values (Figure adapted from** Mohammadi and Hayley (2024a)**)**

Previous experimental programs have identified a consistent pattern in the $\varepsilon - \sigma_v$ curve, marked by a relatively linear trend at

higher pressure ranges. Linear curve fitting (as illustrated in Figure 1.a) allows for the derivation of two parameters: the

coefficient of volume compressibility ($a_0$), corresponding to the slope of this linear segment, and thaw strain parameter ($A_0$),

corresponding to the intercept of the linear portion and strain axis. In the case of the $e - \sigma_v$ curve, a linear semi-logarithmic

trend is identified as more effective in characterizing stress-strain behaviour (Dumais and Konrad, 2023; Mohammadi and

Hayley, 2024a; Nixon and Morgenstern, 1974; Yao et al., 2017). Following the approach proposed for fine-grained soils (i.e.

Dumais and Konrad 2023, 2024), this idealization allows for the derivation of parameters such as the compressibility index of

thawed soil ($C_c^*$), corresponding to the slope of the line, and the threshold thawed void ratio ($e_{th}^*$), corresponding to the thawed



void ratio associated with a threshold vertical effective stress ($\sigma_{v-th}$) at which phase change and expulsion of excess water (if present) conclude and soil's compression as a result of increase in vertical stress initiates, as illustrated in Figure 2.b.

## 2.1 Thaw strain prediction

Several empirical studies have focused on developing correlations between thaw strain and index properties such as water
content and frozen bulk density (Hanna et al., 1983; Luscher and Afifi, 1973; Nelson et al., 1983; Speer et al., 1973). Speer et al. (1973) established a correlation between thaw strain and frozen bulk density based on tests conducted on samples collected along the potential route for a buried warm pipeline, traversing various regions of the Arctic.. Nelson et al. (1983) developed empirical relationships between thaw strain and basic index properties of frozen soils for nine distinct soil groups. Nelson's equation used properties such as moisture content, porosity, and degree of saturation, to allow for thaw strain estimation in
partially saturated soils. Luscher and Afifi (1973) developed bilinear relations between thaw strain (measured at $\sigma_v$ of about 70 to 100 kPa) and frozen bulk density for three types of coarse-grained soils. Ladanyi (1996) derived a correlation between thaw strains measured at 100 kPa and frozen bulk density using test results on silt and sand samples from multiple sites along the Mackenzie River (Andersland and Ladanyi, 2003). Hanna et al. (1983) established relationships for predicting thaw strain from volumetric water content for six groups of soils (Gravel, High Plastic Fines, Low Plastic Soils, Organics, and Peat) using
thaw settlement test results on samples subjected to thaw under a vertical stress of 50 kPa. Table 1 summarizes various established methods for thaw strain estimation and the range of their applicability.

Despite being derived from relatively large datasets, the developed correlations for predicting thaw strain exhibit intrinsic variability, indicating potential limitations in their predictive accuracy and introducing uncertainties into predictions. As shown on Figure 3.a, thaw strain estimates for relatively similar soil types can vary substantially between different methods,
introducing uncertainties into the predictions. Additionally, while some tools use laboratory-measured thaw strains collected under a specific vertical stress, others rely on thaw strain values measured under pressures equivalent to the overburden pressure at each sample's depth, which varies for each sample. As thaw strain depends on the applied stress, particularly for highly compressible soils, the estimated thaw strain obtained using the latter approach lacks a clear stress reference. Moreover, even for tools that provide a clear stress reference for the estimated thaw strain, instructions are not provided for adjusting the
estimation for stresses different from the reference value. In Sec. 5 of this paper, the complied thaw settlement test results are used to critically evaluate the effectiveness of empirical method outlined in Table 1, for predicting thaw strain This evaluation will focus on identifying which methods yield the most accurate predictions for various soil types, taking into account the limitations and uncertainties discussed previously. By systematically comparing the predicted thaw strains with the actual test results, the strengths and weaknesses of each approach will be highlighted, ultimately guiding practitioners in selecting the
most reliable method for thaw strain estimation in their specific applications.



**Table 1***: Summary of existing empirical relations for estimating thaw strain from index properties.*

| Method | Proposed Equation[a] | Experimental Data Used for Method Development | | Relevant USCS symbols |
| --- | --- | --- | --- | --- |
| | | Vertical Stress (kPa) | Soil Type | |
| Speer et al. (1973) | $\varepsilon = 73.6 - 101.8 \ln(1000 \times \rho_f)$ | Overburden pressure at the sample's depth | Mainly fine-grained samples | - |
| Nixon and Ladanyi (1978) | $\varepsilon = 90 - 86.8 \left(\dfrac{\rho_f}{\rho_w} - 1.15\right)^{0.5} \pm 5$ | ~100 kPa | Silts and clayey silts within the interval of $1.2 < \dfrac{\rho_f}{\rho_w} < 2$ | - |
| Ladanyi (1996) | $\varepsilon = 85 - 85 \left(\dfrac{\rho_f}{\rho_w} - 1.1\right)^{0.5} \pm 8$ | ~100 kPa | Silt and sand samples | - |
| Luscher and Afifi (1973) | Bilinear relation between $\varepsilon_{th}$ and $\rho_f$ | 70 to 100 kPa | Soil-specific relation for three sub-group of coarse-grained soils See Figure 3.a | Clean Sand: SW or SP Silt: ML Silty Sand: SM |
| Nelson et al. (1983) | $\varepsilon = A_{n-1}n^2 + A_{n-2}n$ $+ A_{n-3}\dfrac{n^2 w}{S_r}$ $+ A_{n-4}\dfrac{n}{S_r}$ $+ A_{n-5}\dfrac{n}{w}$ $+ A_{n-6}$ | Overburden pressure at the sample's depth | Soil-specific relation for nine sub-group of soils. See Figure 3.a for a visualized version. Charts are created assuming $S_r = 100\%$ and $G_s = 2.7$ (for non-organic) and $G_s = 2.3$ (for organic) | Fat Clay: CH, CI Lean Clay: CL Silty Gravel: GM, GC ML Lowland: MH ML upland: ML Silty Sand: SC, SM Clen Sand: SP, SW Clean Gravel: GP, GW |
| Hanna et al. (1983) | Linear or bilinear relation between $\varepsilon$ and volumetric water content | 50 kPa | Soil-specific relation for five subgroups of soils (See Figure 3.b) | Gravel: GP, GW, SW, GM High Plastic Soils: CI, CH, MH Low Plastic Fines: SP, SC, SM, ML, CL Organic Silt: OL Peat: PT |

a) Refer to Appendix A for the full list of symbols, their meaning, and the unit of each variable.



**Figure 3:** Empirical methods for estimating thaw strain ($\varepsilon$) from (a) frozen bulk density ($\rho_f$) and (b) volumetric water content. The correlations are grouped by major soil type. In each box, the line type represents the reference method, while the colour indicates the names of subgroups (if the major soil types are further divided). Although several methods have been developed using $\rho_f$, only one reference uses volumetric water content. Note the variability of $\varepsilon$ for a given $\rho_f$ across different methods developed in various references for coarse-grained and fine-grained soils.

## 2.2 Formulating strain-stress relationship

Recent studies have focused on characterizing the stress-strain behaviour of thawing permafrost and developing correlations between comparative parameters and index properties. These efforts are crucial for constructing stress-strain relationships that facilitate the determination of thaw strain (or void ratio) as a function of vertical stress, as illustrated in Figure 2.





Dumais and Konrad (2023) developed a framework for thaw consolidation of fine-grained permafrost, providing insights into idealizing stress-strain behaviour of permafrost sediments. They employed the concept of residual stress, defined as the effective stress sustained by thawed soil under undrained conditions, to differentiate between ice-rich and ice-poor permafrost samples. Their findings suggest that a linear semilogarithmic relation between effective stress and void ratio is suitable for ice-poor permafrost, while an idealized bilinear relationship is more appropriate for ice-rich soils. Using thaw settlement test

results from the Canadian Arctic Gas Pipeline Project conducted in the 1970s by Northern Engineering Services Company Limited (1976), they established correlations between index properties and the thaw settlement parameters.

Mohammadi and Hayley (2024a) conducted a comprehensive review of the thaw settlement behavior of coarse-grained permafrost samples. Their analysis incorporated thaw settlement test results from various sources, including Allard et al. (2020) and NESCL (1976, 1977b, a). Using thaw settlement data alongside experimental data on the minimum void ratio for granular

soils, they developed empirical relations for predicting thawed void ratio at a specific effective stress of 15 kPa ($e_{15}^*$). They suggested that, similar to fine-grained soils, a linear semilogarithmic relation between effective stress and the void ratio can be adopted for coarse-grained soils. However, they noted that the $C_c^*$ values for these sediments were small, with a mean value of 0.11, indicating minimal sensitivity of thaw strain to applied stress in these sediments. The empirical tools developed for predicting $e_{15}^*$ relied on properties such as median particle size and coefficient of uniformity.

Despite the prevalence of highly organic permafrost in various regions such as the Hudson Bay Lowlands in Canada, studies on the thaw settlement behavior of organic soils are limited (Wang et al., 2023). Highly organic permafrost is particularly prone to thaw settlement due to its high compressibility and its occurrence in the upper layers of permafrost, which are more susceptible to thawing. Using thaw settlement test results on frozen peat samples sourced from NESCL (1976), Mohammadi and Hayley (2024b) showed that gravimetric water content can effectively predict thaw settlement parameters such as $C_c^*$ and

$e_{th}^*$ (at an assumed threshold vertical stress of 1 kPa) for highly organic soils.

Building upon these studies and to enhance the applicability of the compiled dataset for a deeper understanding of the stress-strain behaviour across various soil types, an idealized stress-strain curve was developed, and comparative parameters were extracted for each test result in the dataset. A detailed comparison of these parameters across different soil types will be presented in Sec. 0, contributing to a more comprehensive understanding of thaw settlement evaluation and informing future

research directions.

## 3 Data aggregation

To develop a comprehensive dataset of thaw settlement data, a thorough review of the literature reporting thaw settlement test results was conducted. Relevant data, including test outcomes, sample properties, and borehole information, were extracted. Thaw settlement test results, such as void ratio and/or thaw strain versus vertical stress, were digitized from figures using

PlotDigitizer software (https://plotdigitizer.com) or extracted from tabulated data in various reports. This same procedure was applied to borehole data and particle size distributions, when available.





In total 446 test results were identified and extracted, primarily from two sources: a testing program by Northern Engineering Service Company Limited (NESCL) as part of the Canadian Arctic Gas Pipeline Project in the Mackenzie Valley (Canadian Arctic Gas Study Limited. and Northern Engineering Services Company., 1977; Northern Engineering Services Company

Limited, 1976, 1977c, a, b, d) and a testing program by the Centre d'Études Nordiques (CEN) on samples collected from various villages in Nunavik, northern Quebec (Allard et al., 2020, 2023; Aubé-michaud et al., 2020b, a; Chiasson et al., 2020a, b; L'Hérault et al., 2014; St-amour et al., 2020).

In addition to these primary sources, test results were gathered from other data resources (indicated by ODS), including test results included in Watson et al. (1973a, b) and tests conducted as part of permafrost hazard mapping for different communities

in Yukon (Benkert et al., 2015c, a, b, 2016). Figure 4 illustrates the locations of the boreholes collected from various data sources.

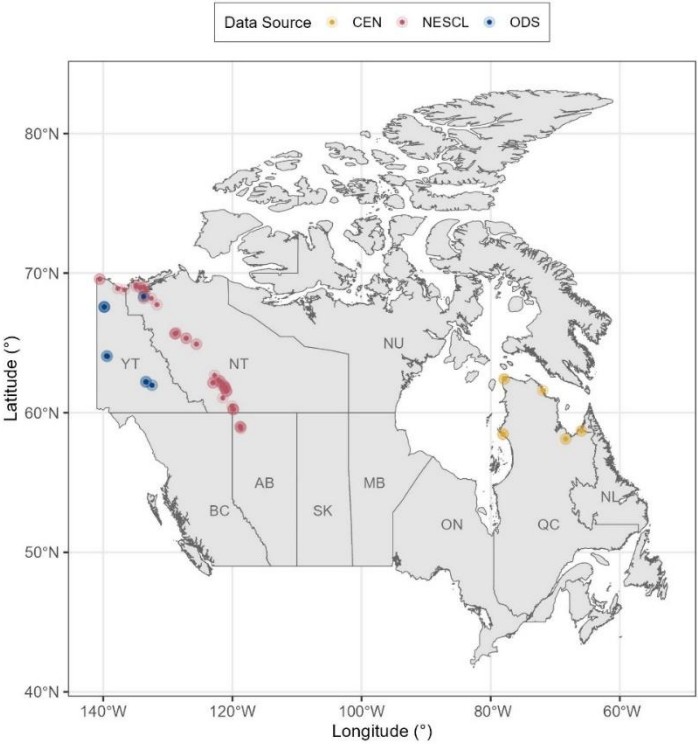

**Figure 4: Locations of boreholes from various sources, including data collected by the Centre d'Études Nordiques (CEN), the Northern Engineering Services Company Limited (NESCL), and other data sources (ODS). The markers are rendered with**
**transparency; areas with higher concentrations of boreholes appear darker. Map created using the *canadianmaps* R package** (Joelle Cayen, 2023)

The collected data underwent thorough evaluation, homogenization, and merging processes to construct a comprehensive dataset, named the Permafrost Thaw Settlement Dataset (PTS), which is publicly available (DOI: 10.5281/zenodo.14538524). The collected data was rigorously examined, avoiding reliance solely on reported sample properties and test results.

Measurements of sample dimensions, mass, and deformation readings during thaw settlement test (if included in the original

sources) were relied upon to derive or re-calculate sample properties and test results. This approach was adapted to add an extra layer of validation and mitigate potential errors in the reported values for these properties or test results in the original reports.

Each sample in the PTS dataset was assigned a unique identifier, referred to as "unique_id," to ensure distinct identification
across various datasets. To standardize the data, all reported values in different unit systems were converted to SI units. A quality control process was implemented using values of 0, 1, and 2 to denote low (e.g. erroneous or insufficient data), moderate (e.g. contradictory properties), and high-quality data, respectively, and each test was flagged accordingly. A detailed description of each variable included the PTS dataset, along with their units of measurement, and relevant notes, is outlined in Appendix A.

While subsets of this data have previously been used to investigate the characteristics of the stress-strain relationship in various permafrost soils (e.g. Dumais and Konrad 2023, 2024; Mohammadi and Hayley 2024b, 2024a), the data used have not been published or made publicly available. This underscores the significance of developing the PTS dataset, which centralizes this data and enhances its accessibility.

### 3.1 Thaw settlement test results

A compilation of thaw settlement test results, including unique sample ID, vertical stress, vertical deformation, thaw strain, void ratio during each loading step is stored in the "thaw_settlement_test_results.csv" file in the PTS dataset. In instances where the sample's deformation and initial height were available, thaw strain and void ratio were calculated based on these measurements. Where deformation was not included in the original reports, and only thaw strain or void ratio was reported, these values were directly included in the dataset. In cases where only one of these two variables (void ratio and thaw strain)
was reported, the missing variable was calculated based on the available variable and sample's properties. This approach was adopted to ensure consistency in the dataset format and to facilitate the extraction of various comparative parameters for each sample.

### 3.1.1 Testing procedure

Due to the lack of a standard procedure for conducting thaw settlement test, the test procedures adopted across the different
sources vary. While the procedure usually involves thawing undisturbed permafrost samples in a modified oedometer cell and measuring the vertical deformation, different loading schemes in terms of the magnitude of the initial load, the number of loading steps, and the range of applied vertical stresses are used across various sources, and even within one testing program. Figure 5 illustrate the variability in testing procedure in terms of minimum and maximum applied vertical stress during each test, and the frequency of the number of loading steps across various data sources in PTS dataset.

For the testing procedure adapted by NESCL, the applied vertical stress during thawing varied between 0.35 kPa and 38.3 kPa, with the majority of samples tested at initial vertical stress between 2.5 kPa and 9.3 kPa (representing 10th and 90th percentile of initial vertical stress, respectively). Most samples were then loaded up to vertical stresses of approximately 66.5 kPa to 77

kPa. However, in a limited number of tests, vertical stresses outside this range were used, with the overall range of maximum applied vertical stress varying between 8.7 kPa and 98.1 kPa. For CEN data, permafrost samples were subjected to thawing
under vertical stress of 25 kPa, followed by two incremental loads at 50 kPa and then 100 kPa, resulting in a total of three deformation readings (L'Hérault et al. 2014).

Most tests from sources other than NESCL, are completed typically with three loading steps. In the case of NESCL data, more than 80% of tests are completed with four loading steps, 6.5% at five loading steps, and limited number of tests at 3, 6, or 7 loading steps. Tests with less than three loading steps were occasionally reported, however, they were excluded when
calculating thaw settlement parameters, and were flagged as "0", indicating tests with insufficient/inaccurate data or sample properties.

 In addition to variation in loading scheme, a variation in sample dimensions exist across different testing programs. The sample diameter for the CEN data was approximately 97 mm, with sample heights ranging from 70 mm to 150 mm, and most samples falling within the 120 mm to 140 mm range. For the NESCL data, a thaw settlement cell with an internal diameter of
63.5 mm and a height of about 50 mm was used. If available, sample or cell dimensions are included in the PTS dataset.

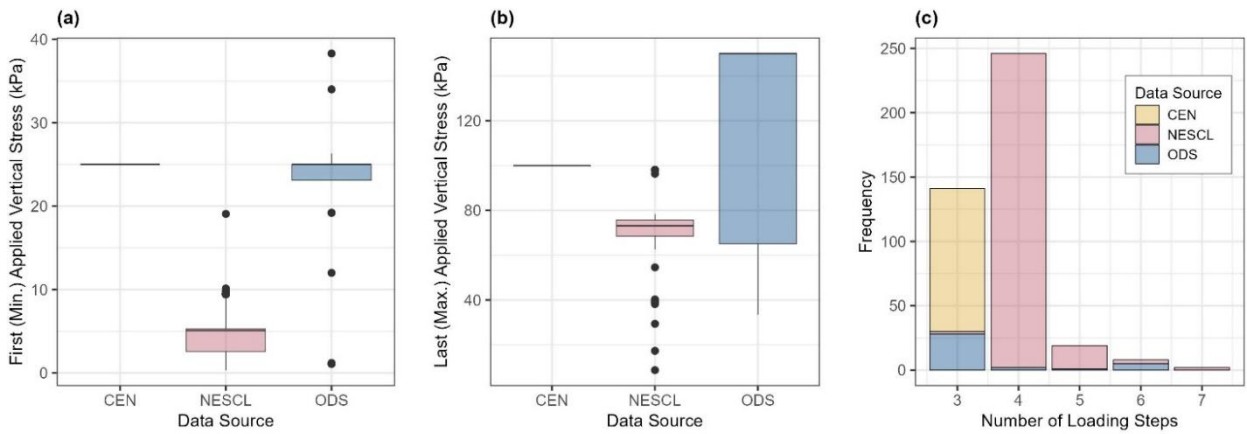

**Figure 5: Variability of test procedure across different data sources (a) a box plot showing the range of minimum vertical stress (applied during first loading step) (b) a box plot showing the range of maximum vertical stress (applied during last loading step), and (c) a bar chart showing the frequency of tests completed with specific number of loading steps across all data sources**

**3.2 Sample information**

**3.2.1 Soil group**

Based on the available information on samples particle size distribution and soil classification/description from the original sources, the samples were categorized into three main groups: Fine-grained, Coarse-grained, and Peat samples. For mineral soils, the Unified Soil Classification System (USCS) criteria were employed to assign soil category. Samples with more than
50% of particles larger than the No. 200 sieve size were categorized as coarse-grained, while those with less were classified as fine-grained. The soil symbols reported in the original sources, when available, are included in the PTS dataset. The supporting document detailing the methodologies used to obtain these symbols is included in Appendix A.

As organic content was reported for only a limited number of samples, a systematic classification of organic samples based on their organic content was not feasible. However, samples labelled as "peat" in the original NESCL reports were presumed to

have significantly high organic content and were classified under the "Peat" soil category in the dataset. Figure 6 presents the relative proportions of different soil categories within the dataset.

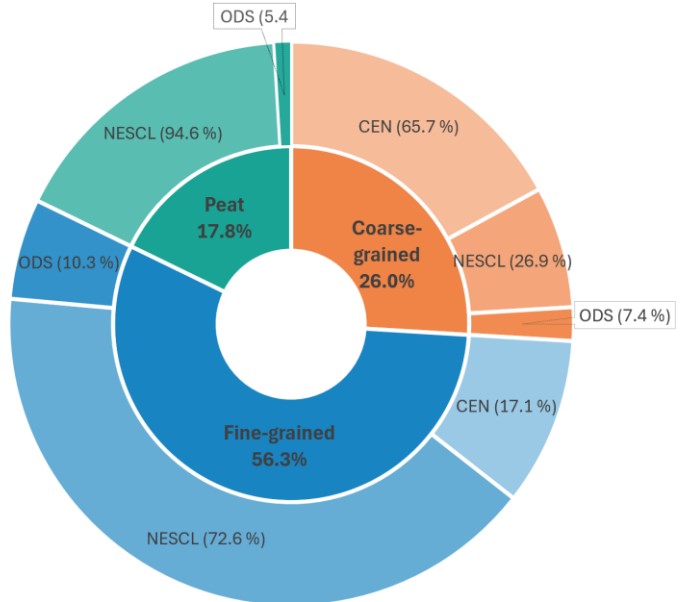

**Figure 6. The relative proportions of three different soil categories within the PTS dataset and their distribution among various sources**

**3.2.2 Index properties**

Information on soil texture, including the fractions of gravel, sand, and fines (and, in some cases, further divided into silt and clay), were among the properties most consistently reported in the original sources and included in PTS dataset. In some instances, Atterberg limits were reported for samples with significant fines content. However, these properties are not uniformly reported across all samples, resulting in incomplete data.

Measurements of sample dimensions and mass (if included in the original sources) were included in the dataset and used to calculate the gravimetric water content and frozen bulk density. If sample measurements were not reported, the reported values for gravimetric water content and frozen bulk density were directly included in the dataset. Gravimetric water content and frozen bulk density were among the most consistently reported properties for the samples. These properties facilitated the calculation of the frozen void ratio ($e_f$) for the samples. The determination of $e_f$ also requires the specific gravity ($G_s$) of the

soil, which was measured for only a limited number of samples. In cases where a measured $G_s$ was not reported for sedimentary samples, specific gravity values ranging from 2.6 to 2.7 were assumed, based on recommendations from the original sources and typical values reported in the literature for sedimentary soils (Andersland and Ladanyi, 2003). For peat samples, a value



of 1.5 was assumed, as recommended by the original data sources. For all the samples, the frozen void ratio ($e_f$) was calculated using the following equation:

$$e_f = \frac{(1+0.01 \times w)G_s \rho_w}{\rho_f} - 1 \tag{1}$$

Where $w$ is the gravimetric water content (in %), $G_s$ is the specific gravity of soil particles (dimensionless), $\rho_w$ is the water density (1000 kg.m$^{-3}$), and $\rho_f$ is the frozen bulk density (in kg/m$^3$).

Properties such as $w$, $\rho_f$, and $e_f$ provide insight into permafrost conditions prior to thaw and serve as proxies for inferring the excess ground ice. These properties are commonly used as predictor variables to develop correlations for thaw strain. Table 2
summarises the descriptive statistics for $w$, $\rho_f$, and $e_f$ across each soil group in the PTS dataset. Figure 7 illustrates the range and variability of these properties within each group, providing a visual comparison of their distribution. The data reveals that peat samples exhibit significantly different values compared to the other soil groups. For all three variables, fine-grained soils display the greatest variation around the mean, as evidenced by the higher coefficient of variation. Frozen bulk density shows less skewness compared to the other variables, with the greatest negative skewness observed in peat samples, followed by fine-
grained and coarse-grained soils. For $w$ and $e_f$, the distribution of data around the mean is more relatively uniform for peat samples but is skewed with a pronounced right tail for fine-grained and coarse-grained soils. Obtained from samples across vast regions and encompassing significant variation, these data are valuable for fitting statistical distributions that best represent the observed variability and shape of the data. This is crucial for accurately modelling uncertainty and conducting probabilistic analyses related to thaw settlement. Moreover, the descriptive statistics can help assign more realistic
deterministic values to these variables based on soil type.

**Table 2: Descriptive statics of index properties including gravimetric water content ($w$), frozen bulk density ($\rho_f$), and frozen void ratio ($e_f$) across three different soil groups in PTSD dataset**

| Variable | Soil Group | No. of Samples | Min. | Max. | Mean | Standard Deviation | COV (%) | Skewness |
|---|---|---|---|---|---|---|---|---|
| $w$ (%) | Coarse-grained | 108 | 15.6 | 120.3 | 42.7 | 23.5 | 55.0 | 1.4 |
| | Fine-grained | 234 | 12.0 | 513.4 | 59.8 | 63.7 | 106.5 | 4.2 |
| | Peat | 74 | 117.6 | 2322.0 | 1006.2 | 499.0 | 49.6 | 0.4 |
| $\rho_f$ (kg/m$^3$) | Coarse-grained | 108 | 1151.0 | 2020.3 | 1677.6 | 212.7 | 12.7 | -0.6 |
| | Fine-grained | 234 | 866.6 | 2087.2 | 1648.1 | 257.7 | 15.6 | -0.7 |
| | Peat | 74 | 608.7 | 1102.1 | 907.9 | 76.3 | 8.4 | -1.2 |
| $e_f$ | Coarse-grained | 108 | 0.52 | 3.95 | 1.36 | 0.75 | 55.1 | 1.3 |
| | Fine-grained | 234 | 0.43 | 17.77 | 1.85 | 2.07 | 111.9 | 4.6 |
| | Peat | 74 | 1.99 | 39.41 | 17.77 | 8.67 | 48.8 | 0.4 |





**Figure 7: Detailed comparison of index properties across three distinct soil groups. The figure includes boxplots that illustrate the distribution of (a) gravimetric water content ($w$), (b) frozen bulk density ($\rho_f$), and (c) frozen void ratio ($e_f$) within each group. Additionally, each box plot is accompanied by histograms for each variable illustrating its frequency distribution of corresponding index properties within each soil group.**



### 3.2.3 Borehole data

Along with the tabulated data, the borehole logs were carefully examined, and relevant information were extracted. The PTS
dataset includes information such as the borehole name, sample depth, descriptions of soil and ground ice at the depths where
samples were collected, all extracted from borehole logs. The supporting documents detailing the methodologies used to
describe soil and ground ice are included in Appendix A.

### 3.3 Particle size distribution

Although the mass fractions of gravel, sand, and fines were consistently reported, detailed particle size distribution data was
available for only a subset of the samples. If available, detailed particle size distribution for the sample was extracted and
stored in the "particle_size_distribution.csv" file within the PTS dataset. Including detailed particle size distribution provides
a comprehensive view of soil texture and enables the derivation of various parameters, thereby enhancing the future application
of the data in diverse contexts.

### 3.4 Metadata and supplementary material

In addition to the sample properties and test results, the metadata, including the location of the boreholes and references to the
original reports containing data relevant to each borehole, is stored in the "borehole_locations_and_sources.csv" file within
the PTS dataset. If the exact coordinate of sampling location or borehole was not available, the approximate location of the
site as described in the main source is included in the metadata.

### 3.5 Idealized stress strain behaviour

Using the methods outlined in Sec. 2.2 for formulating the stress-strain behaviour for different soil types, existing test results
were used to construct an idealized stress-strain curve, which allowed for extraction of thaw settlement parameters for each
test such as the one shown in Figure 2. The idealization allowed for differentiating the ice-rich and ice-poor behaviour during
each test and obtaining void ratio or thaw strain as a function of vertical stress. Additionally, efforts were made to distinguish
the different components of thaw strain during each test including phase change, excess water expulsion and soil skeleton
compression using the idealized behaviour.

### 3.5.1 Thaw settlement parameters

Two parameters ($A_0$ and $a_0$) were derived from $\varepsilon - \sigma_v$ plots using regression analysis. The linear segment of each curve was
identified, and linear regression was applied to data points within this segment. A minimum of three loading steps was required
for the regression. In cases where only three points were available, all were used. However, for tests with more than three
points, an initial regression was performed using all points, and the R-squared ($R^2$) value was checked. If the $R^2$ was
satisfactory (greater than an arbitrary value of 0.98), the model was accepted. Otherwise, points were iteratively removed,





starting with the exclusion of the lowest vertical stress, until an acceptable $R^2$ was achieved or the minimum number of points was reached. In the final regression model, the intercept represented $A_0$, and the slope represented $a_0$.

In the context of void ratio-vertical stress plots, a linear relationship between the void ratio and the logarithm of vertical

(effective) stress is generally identified as the best fit, as established in the literature (Dumais and Konrad, 2023; Nixon and Morgenstern, 1974; Yao et al., 2017). The onset of this linear relationship is marked by a threshold effective stress, commonly known as residual stress for soils with low permeability. Residual stress, an intrinsic soil characteristic, depends on soil texture and the initial ground ice content. It is believed that residual stress for ice-rich samples is relatively small but can be within a physically significant range for ice-poor samples (Dumais and Konrad, 2023; Nixon and Morgenstern, 1973).

To idealize void ratio-effective stress relationship for each sample, linear regression was applied after logarithmically transforming the vertical stress. The resulting slope of this regression model was extracted as $C_c^*$. Then, this regression model was used to differentiate between ice-rich and ice-poor behaviour, and define the residual stress (i.e., the threshold vertical stress) as outlined by Dumais and Konrad (2023). To achieve this vertical stress at the initial thawed void ratio, ($e_i^* = \frac{e_f}{1.09}$), was extrapolated to obtain a hypothetical threshold vertical stress for each sample. If the obtained value was greater than 1

kPa, the sample was classified as ice-poor, and this predicted vertical stress was considered as the threshold vertical stress. Conversely, if the predicted value was unrealistically small (lower than 1 kPa), ice-rich behaviour was assumed, and the threshold vertical stress was set to 1 kPa, representing a reasonably low and physically comprehensible value.

While for coarse-grained soils, the concept of residual stress is less relevant due to their high permeability, to ensure consistency in obtaining comparative parameters across all soil types, the method outlined by Dumais and Konrad (2023) to

obtain residual stress in fine-grained soils was applied to all three soil types.

After defining the threshold vertical stress, the developed regression model was used to predict the threshold thawed void ratio ($e_{th}^*$, as shown in Figure 2.b) for each sample. For ice-poor samples, the threshold thawed void ratio is equal to the initial thawed void ratio. For ice-rich samples, the threshold thawed void ratio is generally lower than the initial thawed void ratio, with the difference representing the excess ice within the sample.

For each sample, comparative parameters such as $A_0$, $a_0$, $C_c^*$, $e_{th}^*$, $\sigma_{v-th}$, and the sample condition (indicating ice-rich or ice-poor behaviour) have been stored in the PTS dataset. These parameters form a basis for future comparisons of thaw settlement characteristics across various soil types. A detailed comparison of these parameters is discussed in Sec. 4.

### 3.5.2 Inferred sample condition (ice-rich versus ice-poor)

Approximately 70% of coarse-grained, 60% of fine-grained, and 70% of peat samples were classified as exhibiting ice-rich

behaviour. This high percentage may be partly due to a sampling and testing bias in thaw settlement tests, where the emphasis is often on ice-rich samples to better represent worst-case scenarios. Figure 8 provides a comparison of the distribution of frozen void ratios in ice-rich and ice-poor samples, along with a histogram showing the occurrence of these two conditions across varying fines fractions and soil classifications, which serve as proxies for soil texture and composition.




The box plots in Figure 8 show that for both coarse-grained and fine-grained soils, the frozen void ratio in ice-poor samples exhibits a relatively narrow range. Additionally, ice-rich and ice-poor conditions are well differentiated by $e_f$, with minimal overlap between the two classes, especially in fine-grained soils. Conversely, the histograms in Figure 8 illustrate that ice-rich and ice-poor behaviour occurs uniformly across various fines fractions and soil classifications, indicating that the presence of excess ice in samples is independent of soil composition based on PTS dataset. However, it is generally understood that fine-grained soils are more prone to higher ground ice content due to the formation of segregated ice.

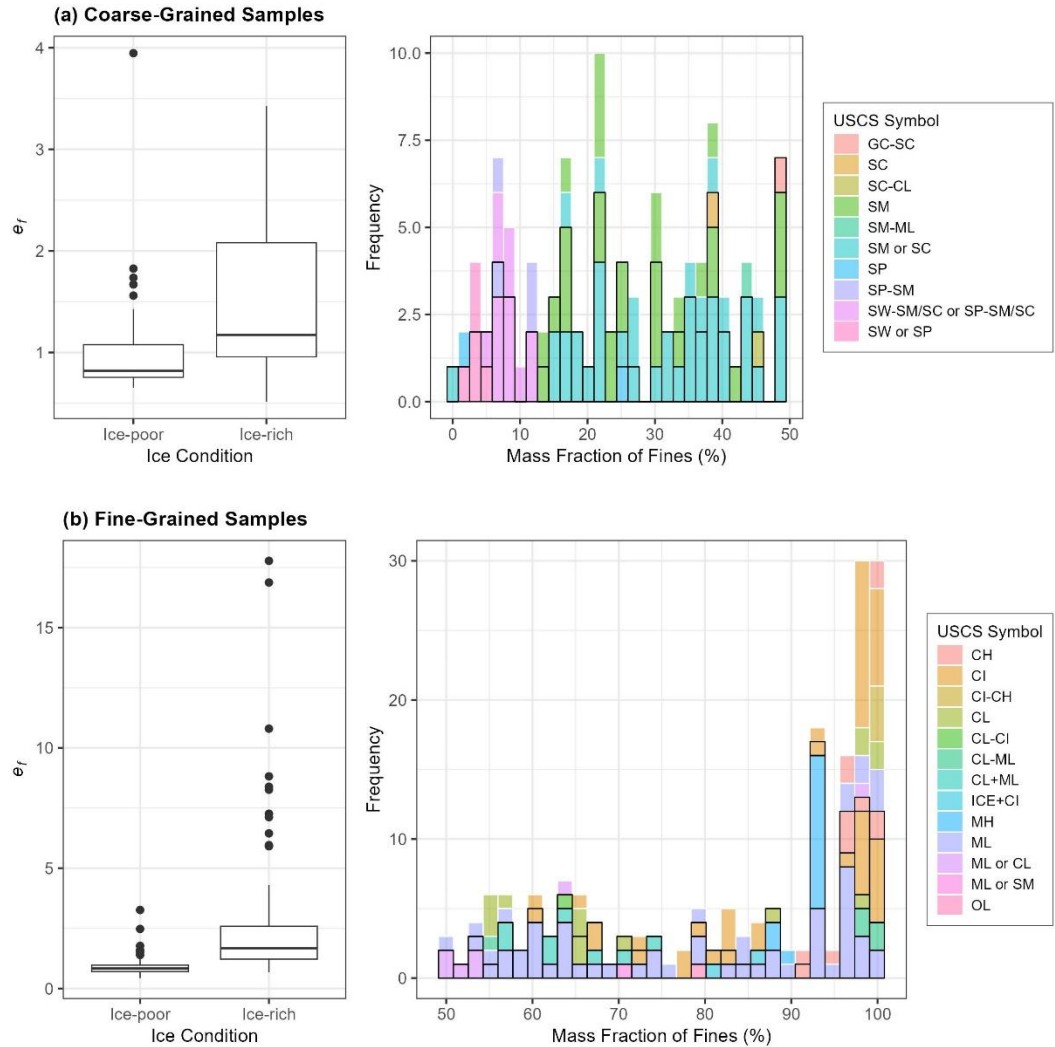


**Figure 8: The boxplots on the left side illustrate the distribution of frozen void ratio ($e_f$) for ice-rich and ice-poor conditions in the PTS dataset for (a) coarse-grained samples and (b) fine-grained samples. The frequency histogram on the right side illustrates the distribution of fines content for (a) coarse-grained and (b) fine-grained samples. The histogram series represent multiple USCS soil symbols (with varying fill colors) and ice conditions (with different border styles) to demonstrate the relationship between soil type**
**and ice condition across the samples.**



### 3.5.3 Thaw strain estimation

The idealized void ratio-vertical stress relationship enables the approximate differentiation of various thaw strain components contributing to the total thaw strain under a given applied vertical stress. Utilizing the idealized $e - \log_{10}(\sigma_v)$ relationship, allows for the derivation of the portion of thaw strain attributed to phase change ($\varepsilon_{pc}$), the expulsion of excess water ($\varepsilon_{ew}$), and soil compression ($\varepsilon_{sc}$) at any given vertical stress. These components can be calculated using the equations summarized in Table 3, based on parameters such as $C_c^*$, $e_{th}^*$, and $\sigma_{v-th}$ and $e_f$. The detailed derivation of these equations is discussed in the appendix A.

**Table 3: Summary of equations used for calculating various components of thaw strain based on $e - \log_{10}(\sigma_v)$ idealization**

| Thaw Strain Component | Equation | Required inputs |
|---|---|---|
| Phase Change ($\varepsilon_{pc}$) | $\varepsilon_{pc}(\%) = \dfrac{e_f - 0.917 \times e_f}{1 + e_f} \times 100$ | $e_f$: Frozen void ratio |
| Expulsion of Excess Water ($\varepsilon_{ew}$) | $\varepsilon_{ew}(\%) = \dfrac{0.917 \times e_f - e_{th}^*}{1 + e_f} \times 100$ | $e_f$: Frozen void ratio<br>$e_{th}^*$: Threshold thawed void ratio |
| Soil Compression ($\varepsilon_{sc}$) | $\varepsilon_{sc}(\%) = \dfrac{C_c^* \times \log_{10}(\frac{\sigma_{v-th}}{\sigma_v})}{1 + e_f} \times 100$ | $e_f$: Frozen void ratio<br>$C_c^*$: Compression index of thawed soil<br>$\sigma_{v-th}$: Threshold vertical stress<br>$\sigma_v$: Vertical stress |

In some thaw settlement tests from the PTS dataset, measured thaw strains were found to be lower than $\varepsilon_{pc}$, indicating that swelling exceeded the anticipated volume reduction due to phase change after thawing. These samples were flagged in the PTS dataset with a binary variable, "swelling_flag," where a TRUE value was assigned to these samples. For most of these samples, swelling occurred only during the initial loading steps; however, in some cases, it persisted even after an additional load was applied to the thawed samples.

Swelling, when it occurred, was detectable only in ice-poor samples. In ice-rich conditions, swelling was not detectable, as it was fully compensated by the volume reduction due to phase change and the expulsion of excess water. Swelling was identified in 79 out of 148 samples classified as ice-poor, with approximately 80% being fine-grained, 14% coarse-grained, and 6% peat samples.

Figure 9 illustrates the distribution of frozen void ratios for swelling and non-swelling samples in ice-poor coarse-grained and fine-grained categories, as well as the occurrence of swelling across varying fines fractions and soil classifications. The boxplots indicate that, in both soil categories, swelling was predominantly observed in denser samples. The histograms demonstrate that swelling was primarily noted in samples with higher fine fractions. For coarse-grained samples, swelling was most commonly observed in silty sands, while in fine-grained samples, it was primarily identified in those with over 90% fines content, classified as high or medium plasticity clay, indicating the dependency of swelling on the soil's composition.



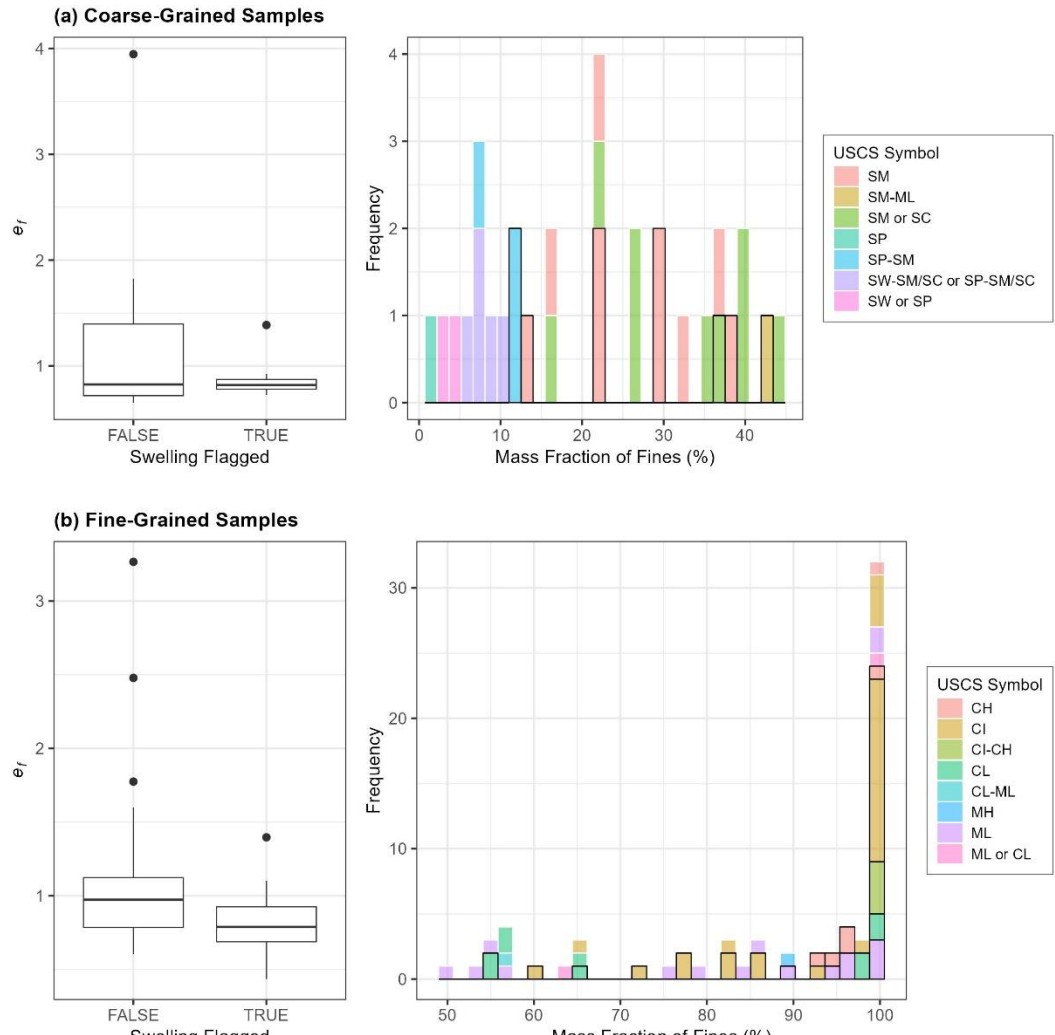


**Figure 9: The boxplots on the left side illustrate the distribution of frozen void ratio ($e_f$) for samples in the PTS dataset, indicative of swelling (TRUE) and no swelling (FALSE) for (a) ice-poor coarse-grained samples and (b) ice-poor fine-grained samples. The frequency histogram on the right side shows the distribution of fines content for (a) ice-poor coarse-grained and (b) ice-poor fine-grained samples. The histogram series represent multiple USCS soil symbols (with varying fill colors) and swelling indication (with**
**different border styles), highlighting the relationship between soil type and swelling occurrence across the samples.**

Using the idealized $e - \log_{10}(\sigma_v)$ relationship, the total thaw strain at any given vertical stress can be calculated by summing the three main components of thaw strain (see Table 3) as follows:

$$\varepsilon_{total}(\%) = \left(\frac{e_f - e_{th}^* + C_c^* \times \log_{10}\left(\frac{\sigma_{v-th}}{\sigma_v}\right)}{1 + e_f}\right) \times 100 \qquad (2)$$

Where $e_f$ is the frozen void ratio, $e_{th}^*$ is threshold thawed void ration, $C_c^*$ is the compression index of thawed soil, $\sigma_{v-th}$ is the

threshold vertical stress, and $\sigma_v$ is the vertical stress.





Alternatively, using the idealized $\varepsilon - \sigma_v$ relationship, the total thaw strain at any given vertical stress can be estimated as follows:

$$\varepsilon_{total}(\%) = A_0 + a_0\sigma_v \tag{3}$$

Where $A_0$ is thaw strain parameter (in %), and $a_0$ is the coefficient of volume compressibility (in % per kPa).

Upon determining the thaw settlement parameters, both Eq. (2) and Eq. (3) can be used to estimate thaw strain at a given vertical stress. However, since the idealized $e - \log_{10}(\sigma_v)$ relationship better captures the overall nonlinear stress-strain behaviour of permafrost samples across a wider range of vertical stresses, Eq. (2) provides more accurate estimates of thaw strain, particularly at lower vertical stress values, compared to Eq. (3).

For each thaw settlement test, various components of thaw strain were calculated and included in the PTS dataset, providing
insight into the contribution of each component to the total thaw strain. A vertical pressure of 100 kPa was selected to determine the portion of thaw strain attributed to soil compression. A visualization of these variables across different soil groups is provided in Appendix A for further details.

## 4 Comparison of thaw settlement parameters across soil categories

Figure 10 depicts the distribution of $A_0$ and $a_0$ values across various soil categories, while Table 4 provides a summary of the
descriptive statistics for these parameters within each category. An analysis of the overall range and mean value of $A_0$ reveals no significant differences between coarse-grained and fine-grained soils, with most samples having $A_0$ values below 30%. However, peat samples exhibit approximately double the mean value compared to other soil types, with $A_0$ value for the majority of samples exceeding 30%.

In coarse-grained and fine-grained soils, Figure 10.a demonstrates a marked disparity in $A_0$ values under ice-poor versus ice-
rich conditions. The mean $A_0$ value for ice-rich samples is notably higher—about four times greater for coarse-grained soils and seven times greater for fine-grained soils—compared to ice-poor samples. Analysis of the 10th and 90th percentiles of $A_0$ reveals minimal overlap in the range of $A_0$ for ice-poor and ice-rich conditions within these two soil categories. The midpoint of the 90th percentile for $A_0$ in ice-poor soils and the 10th percentile for $A_0$ in ice-rich soils serves as a practical threshold for differentiating between these conditions, approximately 12.6% for coarse-grained soils and 13.3% for fine-grained soils. For
peat samples, there is some overlap in $A_0$ values between ice-rich and ice-poor conditions, specifically from 33.4% (the 10th percentile for ice-rich conditions) to 46.6% (the 90th percentile for ice-poor conditions). The mean value of $A_0$ for ice-rich peat samples is almost two times greater than ice-poor samples. The distinction in $A_0$ values between ice-rich and ice-poor conditions, with less evident influence from soil type, highlights the parameter's strong dependence on ground-ice content, rather than soil composition.

Representing the compressibility of the samples, the range of $a_0$ is substantially different across the three soil groups, with notable distinctions between peat samples and mineral soils. Fine-grained samples exhibit a compressibility approximately 1.7 times higher than coarse-grained samples. Peat samples, in contrast, show compressibility values that are 7.1 times higher than




coarse-grained samples and 4.4 times higher than fine-grained samples. For each soil category, the $a_0$ values display only
minor variations in range and mean values between ice-rich and ice-poor samples. The mean compressibility values are
0.047%, 0.076%, and 0.334% per kPa for coarse-grained, fine-grained, and peat samples, respectively. These mean
compressibility values can be used to refine thaw strain estimation through empirical methods, allowing for adjustments based
on the applied vertical stress and soil type.

**Figure 10:** Frequency histograms of (a) the thaw strain parameter ($A_0$) and (b) the coefficient of volume compressibility ($a_0$) across
three main soil groups. Different colors represent the ice conditions, highlighting the lower values of $A_0$ in ice-poor conditions
compared to samples classified as ice-rich. For $a_0$, both ice-rich and ice-poor samples exhibit similar ranges within each soil group;
however, the mean value of $a_0$ is higher for peat compared to mineral soils, and for fine-grained soils compared to coarse-grained
soils.



**Table 4: The descriptive statistics for the thaw strain parameter ($A_0$) and the coefficient of volume compressibility ($a_0$)**

| Parameter | Soil Category | Ice Condition | 10th Percentile | 90th Percentile | Mean | SD | COV |
|---|---|---|---|---|---|---|---|
| $A_0$ | Coarse-grained | Ice-poor | 2.6 | 12.1 | 7.8 | 5.2 | 66.7 |
| | | Ice-rich | 13.1 | 50.2 | 27.6 | 14.4 | 52.2 |
| | Fine-grained | Ice-poor | -1 | 12.6 | 4.6 | 6.1 | 132.6 |
| | | Ice-rich | 13.9 | 52.2 | 32.6 | 15 | 46 |
| | Peat | Ice-poor | -0.4 | 46.6 | 25.5 | 16.2 | 63.5 |
| | | Ice-rich | 33.4 | 58.5 | 45.7 | 10.9 | 23.9 |
| $a_0$ | Coarse-grained | Ice-poor | 0.024 | 0.096 | 0.057 | 0.031 | 54.4 |
| | | Ice-rich | 0.021 | 0.063 | 0.042 | 0.019 | 45.2 |
| | Fine-grained | Ice-poor | 0.029 | 0.114 | 0.076 | 0.033 | 43.4 |
| | | Ice-rich | 0.031 | 0.118 | 0.076 | 0.046 | 60.5 |
| | Peat | Ice-poor | 0.194 | 0.575 | 0.388 | 0.154 | 39.7 |
| | | Ice-rich | 0.167 | 0.439 | 0.312 | 0.175 | 56.1 |

**Note:** These measures are based on a subset of data with definitive $A_0$ and $a_0$ values obtained from linear regression models fitted to test results with $R^2$ values greater than 0.8. Models with $R^2$ values less than 0.8 were excluded, as the corresponding $A_0$ values may be inaccurate due to weak fit.

Figure 11 depicts the distribution of $e_{th}^*$ and $C_c^*$ values across various soil categories, while Table 5 provides a summary of the
descriptive statistics for these parameters within each category. The analysis of $e_{th}^*$ reveals minimal differences between coarse-grained and fine-grained soils, with most samples having $e_{th}^*$ values below 1.5. Peat samples exhibit the highest $e_{th}^*$ values, with a mean value of 13.66, which is 15.5 times higher than coarse-grained soils and 11.4 times higher than fine-grained soils. Unlike the $A_0$ parameter, $e_{th}^*$ does not show a consistent trend when comparing ice-rich and ice-poor conditions. When comparing the $C_c^*$ values, which represents soil compressibility similarly to $a_0$, notable variations are also observed
between soil categories. Peat samples have the highest $C_c^*$ values, with a mean value of 4.73, which is approximately 33.8 times higher than that of coarse-grained soils and 20.6 times higher than fine-grained soils. The coefficient of variation (COV) for $C_c^*$ is highest in coarse-grained and fine-grained soils, indicating greater relative variability, while peat samples exhibit moderate variability. A comparison between ice-rich and ice-poor samples reveals no specific trends in $C_c^*$ based on ice content, underscoring the parameter's dependence on soil composition rather than ice conditions.





**Figure 11:** Frequency histograms of (a) threshold thawed void ratio ($e^*_{th}$) and (b) compressibility index of thawed soil ($C^*_c$) across three main soil categories. Different colors represent the ice conditions.

**Table 5:** The descriptive statistics of $e^*_{th}$ and $C^*_c$

| Parameter | Soil Category | 10th Percentile | 90th Percentile | Mean | SD | COV |
|-----------|---------------|-----------------|-----------------|------|------|------|
| $e^*_{th}$ | Coarse-grained | 0.48 | 1.34 | 0.88 | 0.44 | 50.0 |
| | Fine-grained | 0.63 | 1.68 | 1.20 | 1.03 | 85.8 |
| | Peat | 5.39 | 22.78 | 13.66 | 6.81 | 49.9 |
| $C^*_c$ | Coarse-grained | 0.05 | 0.23 | 0.14 | 0.12 | 85.7 |
| | Fine-grained | 0.06 | 0.43 | 0.23 | 0.29 | 126.1 |
| | Peat | 1.36 | 9.18 | 4.73 | 3.28 | 69.3 |





## 5 Validation of existing thaw strain prediction tools


Empirical correlations relating thaw strain to properties such as water content and frozen bulk density of permafrost samples have been widely used to estimate thaw settlement (Luscher and Afifi, 1973; Nelson et al., 1983; Speer et al., 1973; Watson et al., 1973a). These correlations offer a straightforward and cost-effective approach to estimating anticipated thaw strain upon thawing. However, estimations made using these methods exhibit inherent variability and significant scatter, despite using

relatively large datasets to establish the correlations. Therefore, thaw strain estimates can vary substantially between different methods, introducing uncertainties into the predictions.

In this study, compiled data was used to assess the effectiveness of various existing methods and compare their performance in predicting thaw strain. To achieve this, thaw strain values estimated using the methods outlined in Table 1 were compared with corresponding measured values for each sample in the PTS dataset. The performance of each method in comparison to

others was evaluated using measures such as the Root-Mean-Square Deviation (RMSD), and bias between estimated and actual thaw strain values. RMSD, a measure of average deviation between estimated and actual values, and bias, a measure of systematic deviation from the true values, both provide insights into the accuracy and reliability of predictive models. A lower RMSD indicates better agreement between estimated and actual values, while a bias closer to zero suggests that the method is not systematically overestimating or underestimating thaw strain. Table 6 summarizes the obtained metrics for each method

across major soil categories, while Table 7 presents the metrics for methods that provide soil-specific correlations. Figure 12 visualizes this evaluation by presenting the predicted versus actual thaw strain for various empirical methods.

In this evaluation, careful attention was given to using only a subset of compiled data that fit the applicability range of each method, or each sub-group of soils (if applicable) in terms of soil type. Properties such as soil texture, plasticity limits and USCS classification (if available) were used to identify samples within the PTS dataset that matched each method or subgroup.

To ensure comparability between actual thaw strain and estimated values in relation to vertical stress, thaw strain corresponding to the vertical stress used to derive the correlations for each method was selected (see Table 1). For instance, for the correlations proposed by Ladanyi (1996) and Nixon and Ladanyi (1978), thaw strain measurements at 100 kPa were calculated for each test using Eq. (2) and used as the actual thaw strain for validation against predictions made by these methods. Similarly, the correlations by Hanna et al. (1983) were evaluated using thaw strain at 50 kPa for each soil type. The correlations by Luscher

and Afifi (1973) were developed using thaw strain at vertical stresses ranging from 70 kPa to 100 kPa; therefore, thaw strain values at 100 kPa was used to obtain the metrics presented in Table 6 and Table 7. Both, Hanna's method and the Luscher and Afifi method, provide soil-specific correlations. In both cases, appropriate thaw strain correlations were applied after assigning each sample in the PTS dataset to one of the defined soil subgroups.

In the cases of Speer et al. (1973) and Nelson et al. (1983), thaw strains were not measured at a unique vertical stress; instead,

they were measured at various vertical stresses, selected to replicate the overburden pressure at the sample depths. Therefore, during the validation process, the overburden pressure for each sample was first estimated using an approximate borehole stratigraphy constructed based on the available data. Then, thaw strain values were calculated at these estimated overburden





pressures, using Eq. (2), and were compared with predictions from Speer's and Nelson's methods. Speer's method was applied to all samples classified as fine-grained or coarse-grained in the PTS dataset. For Nelson's method, which provides soil-specific correlations, the appropriate correlations were applied after assigning each sample in the PTS dataset to one of the defined subgroups. Most samples were categorized into Fat Clay, Lean Clay, and ML upland/lowland subgroups, while fewer samples were classified into subgroups such as Clean Sand, Gravel, Silty Gravel, and Organics.

As shown Table 6, among the existing correlations, the method proposed by Nixon and Ladanyi (1978) produced the lowest RMSD and bias when applied to both fine-grained and coarse-grained soils. The RMSD was significantly lower for fine-grained samples, likely due to the development of the original correlations mainly based on thaw strain measurements of silts and clayey silts. A correlation for peat samples was provided solely by Hanna et al. (1983); however, this correlation does not fit the peat sample data well, significantly underestimating thaw strain, as indicated by a high RMSD of approximately 22.66% and a bias of about -18.18%.

As shown in Table 6, a comparison of correlations developed for subgroups of coarse-grained soils by Nelson et al. (1983) and Luscher and Afifi (1973) reveals that, in general, Nelson's correlations resulted in lower RMSD, and bias compared to those of Luscher and Afifi. However, the correlations developed by Nixon and Ladanyi (1978) achieved the lowest RMSD and bias for both clean sand and silty/clayey sands. Although the number of gravel samples within the PTS dataset was limited, the correlation developed by Hannan et al. (1983) was found to be the most effective for this group, followed by the correlation proposed by Nixon and Ladanyi (1978). For sub-groups of fine-grained soils, correlations developed by Nelson et al. (1983) and Hanna et al. (1983) generally exhibited higher RMSD and bias compared to the Nixon and Ladanyi (1978) method, making the latter more effective for predicting thaw strain for different type of these soils. The correlations developed for organic soils are limited, as is the number of samples assigned to the organic subgroup in the PTSD dataset. However, among the methods proposed by Nelson and Hannan, the Nelson yielded lower RMSD and bias for organic soils.

It also should be noted that, in general, correlations developed between thaw strain and frozen bulk density, such as Speer et al. (1973), Nixon and Ladanyi (1978), and Ladanyi (1996) showed comparable performance in terms of predicting thaw strain, with Nixon and Ladanyi (1978) marginally outperforming the other two correlation, indicated by a smaller RMSD and a smaller absolute bias. Moreover, the positive bias value for this method, which indicates an overestimation of thaw strain, is more favourable as it results in conservative estimations. The correlations developed by Hanna et al. (1983) are the recommended tools for predicting thaw strain in Canadian Foundation Engineering Manual (CGS 2023), however the result of this evaluation suggests that other correlations may outperform these correlations, particularly in terms of bias, as this method has a significantly large negative bias, indicative of an underestimation of thaw stain.



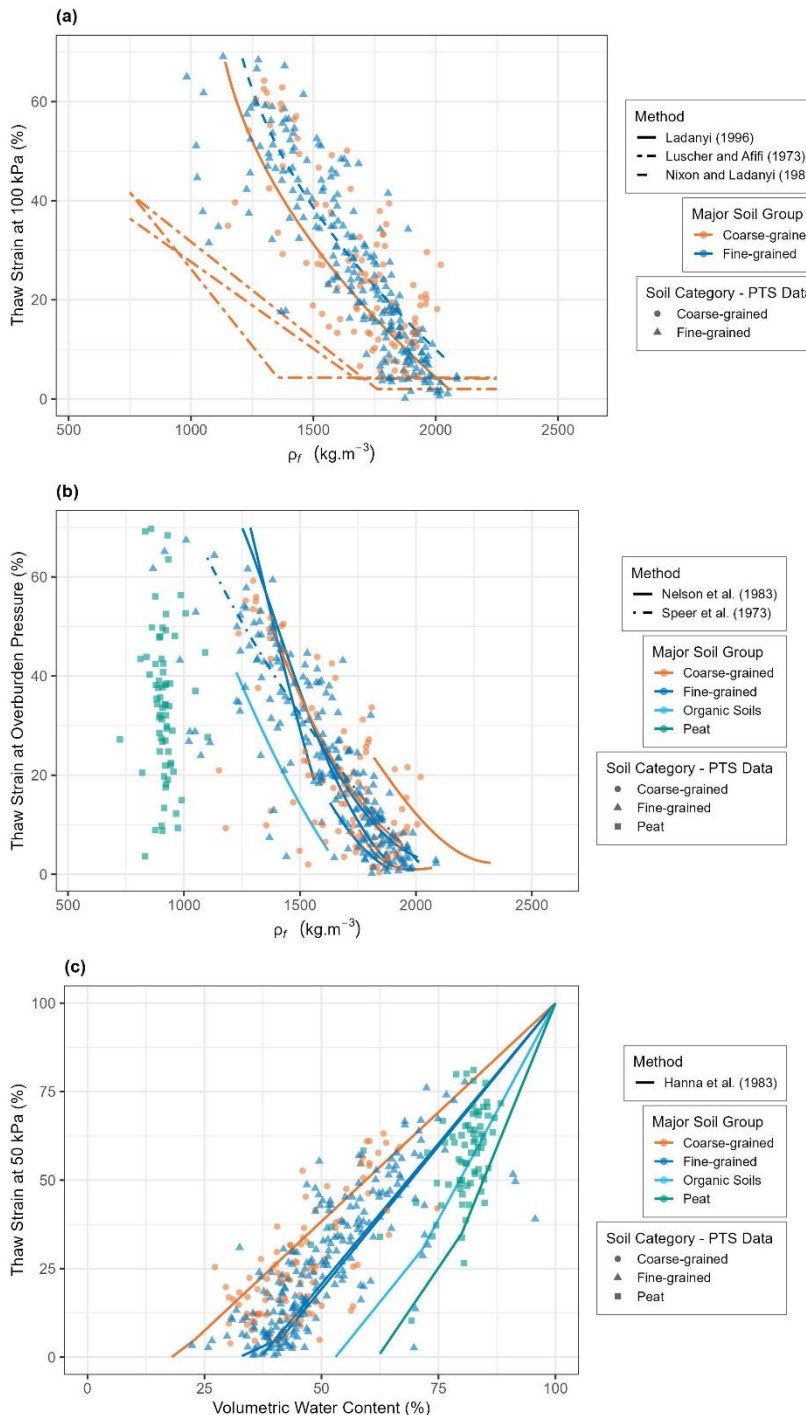

**Figure 12: Visual comparison of empirical tools for predicting thaw strain against experimental data from the PTS dataset. Lines represent the correlations, while points depict the experimental data. Color represents major soil groups both for correlations and** 545 **experimental data. The subplots include: (a) three methods, including Luscher & Afifi (1973),** Nixon and Ladanyi (1978)**, and Ladanyi (1996); (b) two methods, including Nelson et al. (1983) and Speer et al. (1973); and (c) the Hannan et al. (1983) method.**



**Table 6: Root mean squared deviation (RMSD) and bias for thaw strain measurements from PTS dataset and predictions made using different empirical methods**

| Soil Group | Method | Subset size | RMSD | Bias | Note |
|---|---|---|---|---|---|
| Coarse-grained | Hannan et al. (1983) | 107 | 15.18 | -9.99 | See Table 7 for various subgroups |
| | Ladanyi (1996) | 107 | 12.03 | -5.07 | |
| | Luscher & Afifi (1973) | 106 | 23.16 | -19.46 | See Table 7 for various subgroups |
| | Nelson et al. (1983) | 107 | 12.75 | -4.62 | See Table 7 for various subgroups |
| | Nixon and Ladanyi (1978) | 107 | 11.75 | 1.89 | |
| | Speer et al. (1973) | 107 | 12.36 | 2.37 | |
| Fine-grained | Hannan et al. (1983) | 234 | 13.18 | -5.5 | See Table 7 for various subgroups |
| | Ladanyi (1996) | 224 | 10.17 | -4.48 | |
| | Luscher & Afifi (1973) | 95 | 19.22 | -25.05 | See Table 7 for various subgroups |
| | Nelson et al. (1983) | 234 | 16.19 | -3.88 | See Table 7 for various subgroups |
| | Nixon and Ladanyi (1978) | 222 | 8.78 | 2.14 | |
| | Speer et al. (1973) | 234 | 10.65 | 3.15 | |
| Peat | Hannan et al. (1983) | 68 | 22.66 | -18.18 | |

**Table 7: Root mean squared deviation (RMSD) and bias for thaw strain measurements from PTS dataset and predictions made using different empirical methods for various sub-groups of soils**

| Method | Subgroups | Subset size[a] | RMSD | Bias |
|---|---|---|---|---|
| Hannan et al. (1983) | Peat | 68 | 22.66 | -18.18 |
| | Organic Silt | 8 | 28 | -16.15 |
| | High Plastic Soil | 117 | 11.28 | -3.92 |
| | Low Plastic Fines | 214 | 14.56 | -8.39 |
| | Gravel | 7 | 6.59 | 4.76 |
| Luscher & Afifi (1973) | Silt | 96 | 30.15 | -25.09 |
| | Silty Sand | 98 | 23.67 | -19.71 |
| | Clean Sand | 8 | 17.58 | -16.38 |
| Nelson et al. (1983) | Organics | 8 | 13.75 | -1.12 |
| | Fat Clay | 102 | 11.93 | -8.53 |
| | Lean Clay | 33 | 10.94 | -3.64 |
| | Silt | 95 | 21.06 | 1.06 |
| | Silty Sand | 98 | 12.18 | -3.51 |
| | Clean Sand | 8 | 18.69 | -17.87 |

a) Subgroups with subset size of 5 or less are excluded.



## 6 Uncertainty

The data used in this study were collected from a range of published sources rather than generated by the author. As a result,
several sources of uncertainty are inherent in the dataset. One significant source stems from potential errors in the original sampling, testing, and reporting processes. Ideally, undisturbed permafrost samples should be obtained for thaw settlement tests, but in practice, thermal disturbance and water loss can occur during sample collection, transport, and handling. These factors introduce inaccuracies in the measurements and, subsequently, the test results. Additionally, the lack of a standardized protocol for conducting thaw settlement tests across different studies contributes to variability. Each source may have used
different procedures, leading to inconsistent results that add another layer of uncertainty.

To add a layer of validation to the reported test results, in cases where the original test results were provided, actual deformation and sample measurements were used to verify the reported values for thaw strain and void ratio. Discrepancies were occasionally observed, and where necessary, recalculated values based on the original measurements were prioritized to improve data reliability. Additionally, a quality control process was also implemented to flag the reliability of the data. Values
of 0, 1, and 2 were used to denote low, moderate, and high-quality data, respectively. Low-quality data generally represented erroneous or insufficient information, while moderate-quality data often had contradictory properties. High-quality data represented results with minimal inconsistencies.

To maintain consistency across the dataset, all reported values from various sources were converted to SI units, and a unique index was assigned to each sample for easy identification. In cases where data were incomplete or missing, such as when a
specific property was not measured or reported by a source, the missing values were denoted as "NA." This approach was consistently applied across the dataset to ensure that data from different sources could be integrated smoothly. Additionally, if a variable was reported in one source but absent from another, it was still listed in the corresponding data table, with "NA" filling the gaps for missing entries. Certain thaw settlement parameters could not be calculated for some of the test results due to insufficient number of loading steps, which made regression analysis unfeasible. These samples were still included in the
dataset but had "NA" assigned to the thaw settlement parameters.

Despite these uncertainties, the dataset provides a valuable foundation for understanding thaw settlement behaviour. The steps taken to standardize, verify, and flag data inconsistencies help to mitigate some of the uncertainties inherent in the collected sources, ensuring that the dataset remains robust for further analysis.

## 7 Data availability

The original measurements and readings, as extracted from the original sources, are included in a GitHub repository. The processed and homogenized data, reduced to the necessary variables, are published as the Permafrost Thaw Settlement (PTS) dataset, supporting research on permafrost thaw settlement. The PTS dataset is publicly available on Zenodo: [https://doi.org/10.5281/zenodo.14538524](https://doi.org/10.5281/zenodo.14538524)  (Mohammadi and Hayley, 2024c).



## 8 Code availability

The code, written in R (R Core Team, 2023), used for this analysis is available at
https://github.com/Zucchii/ThawSettlement_DataPaper, specifically at commit 5678ccd. This version was used to generate all
scripts, figures, and datasets. The repository also contains links to the associated dataset on Zenodo.

## 9 Conclusions

Thaw settlement remains a significant concern for infrastructure stability in permafrost regions, and with the accelerating
effects of climate change, it is increasingly crucial to improve our understanding and predictive capabilities regarding this
phenomenon. The comprehensive dataset presented in this study, compiled from various published thaw settlement tests,
addresses both the scarcity of data and the lack of standardized procedures for conducting such tests. By aggregating and
standardizing data from different sources, this study provides a valuable resource for cross-comparing thaw settlement
properties across a wide range of permafrost soil types, including fine-grained, coarse-grained, and highly organic soils.

The dataset also contributes to enhancing existing empirical tools used to estimate thaw strain from easily measurable index
properties. An evaluation of some existing tools using the experimental data in the PTS dataset highlighted the stronger
predictive performance of the correlation developed by Nixon and Ladanyi (1978) for estimating thaw strain based on frozen
bulk density. Additionally, comparative thaw settlement parameters were derived from idealized stress-strain curves, offering
insights into the stress-strain behaviour of thawing permafrost. These curves are particularly useful for more accurate

modelling of the thaw settlement process and for adjusting thaw strain estimates under various vertical stresses.

Beyond its immediate applications, the dataset serves as a foundation for future research. It can be used either independently
or in conjunction with localized site data to develop new empirical tools or to refine the precision of existing models. The
standardized and open-access nature of the dataset further supports the broader scientific community in its efforts to study
thaw settlement and its impact on infrastructure.

In conclusion, this study contributes a critical resource to the field of permafrost research by offering a homogenized dataset
that enhances our understanding of thaw settlement behaviour across different permafrost sediments. The findings are expected
to support more accurate predictions of thaw settlement and contribute to the design and maintenance of resilient infrastructure
in permafrost regions, particularly in the face of climate change.

## 10 Author contribution

Zakieh Mohammadi and Jocelyn Hayley conceptualized the study. Zakieh Mohammadi curated the data, performed the formal
analysis, developed the methodology, and prepared the original draft of the manuscript. Jocelyn Hayley acquired the funding,
supervised the project, and reviewed and edited the manuscript.





## 11 Competing interests

The authors declare that they have no conflict of interest.

## 12 Funding Statement

This research was funded by PermafrostNet, supported by the Natural Sciences and Engineering Research Council of Canada (NSERC) Strategic Partnership Grant for Networks (Funding Reference No. NETGP 523228-18).

## 13 Acknowledgements

We thank the original data collectors and institutions whose efforts made this dataset possible, including the Arctic Gas Pipeline project (1970s), the Centre for Nordic Studies at Université Laval, and the Yukon Research Centre at Yukon College.



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





# Appendix A: Supplementary Information and Data


Zakieh Mohammadi[1], Jocelyn L. Hayley[1]

[1] Department of Civil Engineering, University of Calgary, Calgary, T2N 1N4, Canada

Correspondence to: Zakieh Mohammadi (seyedehzakieh.mohamm@ucalgary.ca)



**List of Symbols and Nomenclature in the Manuscript**

| Symbol | Represents | Unit |
|---|---|---|
| $COV$ | Coefficient of Variation | % |
| $C_U$ | Coefficient of uniformity | 1 |
| $a_0$ | Coefficient of volume compressibility | % per kPa |
| $C_c^*$ | Compression index of thawed soil | 1 |
| $S_r$ | Degree of saturation | % |
| $\rho_w$ | Density of water | kg/m$^3$ |
| $\rho_f$ | Frozen bulk density | kg/m$^3$ |
| $e_f$ | Frozen void ratio | 1 |
| $w$ | Gravimetric water content | % |
| $e_i^*$ | Initial thawed void ratio | 1 |
| $D_{50}$ | Median particle size | mm |
| $n$ | Porosity | 1 |
| $\sigma'_0$ | Residual stress | kPa |
| $RMSD$ | Root Mean Square Deviation | Same as the unit of data |
| $H_f$ | Sample height at the frozen state | mm |
| $H_{\sigma_v}^*$ | Sample height at the thawed state at a vertical stress of $\sigma_v$ | mm |
| $SD$ | Standard deviation | Same as the unit of data |
| $A_{n-i}$ | Soil-specific coefficients for the Nelson's method | Unitless |
| $G_s$ | Soil's specific gravity | 1 |
| $\varepsilon_{\sigma_v}^*$ | Thaw strain at a vertical stress of $\sigma_v$ | % |
| $A_0$ | Thaw strain parameter | % |
| $\varepsilon_{pc}$ | Thaw strain component attributed to phase change | % |
| $\varepsilon_{ew}$ | Thaw strain component attributed to expulsion of excess water | % |
| $\varepsilon_{sc}$ | Thaw strain component attributed to soil compression | % |
| $\sigma_{v-th}$ | Threshold vertical stress | kPa |
| $e_{th}^*$ | Threshold thawed void ratio | 1 |
| $e_{\sigma_v}^*$ | Thawed void ratio at a vertical stress of $\sigma_v$ | 1 |
| $\theta$ | Volumetric water content | % |
| $\sigma_v$ | Vertical stress | kPa |

**Note:** In the symbols, an asterisk (*) as a superscript denotes the thawed state, the letter *f* as a subscript indicates the frozen
state, and *th* as a subscript signifies threshold values.



**An Overview of the Permafrost Thaw Settlement (PTS) Dataset**

The PTS dataset comprises several files, each serving a distinct purpose:

- **borehole_locations_and_sources.csv**: This file contains geographic coordinates and source references for test results on samples collected from each borehole location.
- **particle_size_distribution.csv**: This file contains particle size distribution for samples, provided it was included in the original sources.
- **thaw_settlement_test_results.csv**: This file contains the results of thaw settlement tests conducted on various permafrost samples. Each row corresponds to a one loading step on a sample, detailing the applied vertical stress, deformation measurements, strain, and void ratio during that specific loading step.
- **sample_details_with_derived_parameters.csv**: This file provides detailed properties of each sample, including both basic properties and comparative parameters derived from the test results. It serves as a comprehensive summary of the physical and mechanical characteristics of the samples.

For a detailed description of the variables included in the PTS dataset, please refer to the accompanying README.txt file. This document provides comprehensive explanations of each variable, along with their corresponding units.

The variable names in the PTS dataset follow a structured naming convention, incorporating descriptors as outlined in Table A.1. These descriptors help distinguish between various types of values, such as initial and final measurements, reported and calculated values (derived from analyses conducted by the author), assumed values, and typical values.

**Table A. 1: Descriptors added to variable names in the PTS dataset**

| Descriptor | Represents | Example of variable names | Explanation |
|---|---|---|---|
| init | Initial | gravimetric_wc_init, sample_volume_init | Used to mark the initial properties of the sample prior to the thaw settlement test. |
| ave | Average | sample_diameter_ave_or_typ | Represents the average value calculated from multiple measurements. |
| fin | Final | gravimetric_wc_fin | Used to mark the properties measured after the completion of the thaw settlement test. |
| rep | Reported | frozen_bulk_density_rep, | Values included as they were reported in the original sources, in contrast to those calculated or re-calculated in this study. |
| asm | Assumed | solid_grain_density_rep_or_asm | Used when a value was assumed for a property rather than measured. |
| calc | Calculated | frozen_dry_density_calc, frozen_void_ratio_calc | Values that have been calculated based on measurements or other input data. |
| typ | Typical | sample_height_ave_or_typ, sample_diameter_ave_or_typ | Marks typical values, such as the typical sample diameter, which corresponds to the diameter of the odometer cell rather than the measured value for each sample. |



**Visual Description of Soil and Ground Ice**

The Unified Soil Classification System (USCS) or modified adaptations of this system were used to visually assess and classify samples in the original sources. Specifically, an adaptation of the USCS was employed to classify and describe samples in the NESCL reports. For detailed procedures regarding the description of soil samples, refer to the Geotechnical Data Report for the Proposed Arctic Gas Pipeline - Cross Delta Alternative (NESCL 1975).

For data sourced from NESCL reports, the ground ice descriptions were extracted from borehole logs and incorporated into the PTS dataset. This classification adhered to the NRC 7576 guidelines, categorizing ground ice into three major classes based on visual examination: non-visible (N class), visible (V class), and visible ice greater than one inch thick (ICE class). Table A.4 outlines the subgroup symbols used within each category along with their general descriptions as specified in the NRC 7576 guidelines.

**Table A. 2: Ground ice classification symbols and descriptions based on visual examination following the NRC 7576 guideline**

| Category | Group Symbol | Subgroup Symbol | Description |
|---|---|---|---|
| Non-Visible | N | Nf | Poorly bonded or friable frozen soil. |
| | | Nbn | Well bounded frozen soil with no excess ice. |
| | | Nbe | Well bounded frozen soil with excess ice. Free water present when thawed. |
| Visible ice less than one inch tick | V | Vx | Individual ice crystals or inclusions. |
| | | Vc | Ice coatings on particles. |
| | | Vr | Random or irregularly oriented ice formations. |
| | | Vs | Stratified or distinctly oriented ice formation. |
| Visible ice greater than one inch thick | ICE | ICE + "Soil Type" | Ice greater than one inch thick with soil inclusions. |
| | | ICE | Ice greater than one inch thick without soil inclusions. |

**Homogenizing Thaw Settlement Test Results in the PTS Dataset**

The test results in the original sources were reported in various formats. To homogenize these results, the following procedures were applied.





In cases where both the initial sample height and vertical deformation during each loading step were provided, thaw strain and
void ratio were calculated using the following equations:

$$\varepsilon_{\sigma_v}^* = \frac{H_f - H_{\sigma_v}^*}{H_f} \times 100 \qquad (A.1)$$

$$e_{\sigma_v}^* = e_f - \varepsilon_{\sigma_v}^*(1 + e_f) \qquad (A.2)$$

Where $H_f$ is the initial sample height at frozen state, and $H_{\sigma_v}^*$ is the height of thawed sample at vertical stress of $\sigma_v$.

In instances where only the void ratios or thaw strains at each vertical stress were reported, the missing variable for each
loading step was derived using the following equation:

$$\varepsilon_{\sigma_v}^* = \frac{e_f - e_{\sigma_v}^*}{1 + e_f} \qquad (A.3)$$

Where $\varepsilon_{\sigma_v}^*$ is thaw strain measured at vertical stress of $\sigma_v$, $e_{\sigma_v}^*$ is thawed void ratio at vertical stress of $\sigma_v$, and $e_f$ is the frozen
void ratio.

To incorporate this information into the dataset, a variable, ***reported_test_result_type***, was added to the PTS dataset. This
variable differentiates between values that were directly reported and those derived through conversion from one variable to
another. The following values were used to differentiate between various cases:

- **delta_h-sigma_v**: Deformation measurements and initial sample dimensions are available; used to obtain thaw strain
  and void ratio for each loading step.
- **e-sigma_v**: Test results were reported as void ratio against vertical stress; Eq. (A.2) was applied to derive thaw strain
based on void ratios.
- **strain-sigma_v**: Test results were reported as thaw strain against vertical stress; Eq. (A.3) was applied to derive void
  ratio based on thaw strain.
- **e-sigma_v & strain-sigma_v**: Both void ratio and thaw strain at each loading step were reported in the original
  sources. Checks were conducted to ensure compatibility; in the case of discrepancies, test result was flagged.

**Derivations of Various Thaw Stain Components**

The idealized void ratio-vertical stress relationship (as depicted in Figure 2.b) was used to differentiate various thaw strain
components, such as phase change, drainage of excess water, and compression of thawed soil, all contributing to the total thaw
strain under a given applied vertical stress.

The portion of thaw strain attributed to phase change ($\varepsilon_{pc}$) can be calculated using the following equation:

$$\varepsilon_{pc}(\%) = \frac{e_f - e_i^*}{1 + e_f} \times 100 \qquad (A.4)$$

Where $e_f$ is the frozen void ratio, and $e_i^*$ denotes the initial thawed void ratio. Given that $e_i^* = \frac{e_f}{1.09}$, Eq. (2) can be simplified
to:



$$\varepsilon_{pc}(\%) = \frac{e_f - 0.917 \times e_f}{1 + e_f} \times 100 \tag{A.5}$$

The portion of thaw strain attributed to the expulsion of excess water ($\varepsilon_{ew}$) can be estimated using following equation:

$$\varepsilon_{ew}(\%) = \frac{e_i^* - e_{th}^*}{1 + e_f} \times 100 \tag{A.6}$$

Where $e_f$ is the frozen void ratio, and $e_i^*$ denotes the initial void ratio after thawing. Given that $e_i^* = \frac{e_f}{1.09}$, Eq. (4) can be simplified to:

$$\varepsilon_{ew}(\%) = \frac{0.917 \times e_f - e_{th}^*}{1 + e_f} \times 100 \tag{A.7}$$

Since, for ice-poor samples, $e_i^* = e_{th}^*$, this portion is zero for the samples classified as ice-poor.

The portion of thaw strain attributed to soil compression ($\varepsilon_{sc}$) at any given vertical stress can be calculated using the following equation:

$$\varepsilon_{sc}(\%) = \frac{C_c^* \times \log_{10}(\frac{\sigma_{v-th}}{\sigma_v})}{1 + e_f} \times 100 \tag{A.8}$$

Where $e_f$ is the frozen void ratio, and $C_c^*$ denotes the compression index of thawed soil, $\sigma_{v-th}$ is the threshold vertical stress marking the initiation of soil's compression during each test, and $\sigma_v$ is the vertical stress. For each thaw settlement test, various

components of thaw strain were calculated to provide insight into the contribution of each to the total thaw strain. A vertical pressure of 100 kPa was applied to determine the portion of thaw strain attributed to soil compression and total thaw strain. Figure A.1 presents box plots of each component and the total thaw strain for two groups of samples—ice-poor and ice-rich—while comparing three main soil groups.



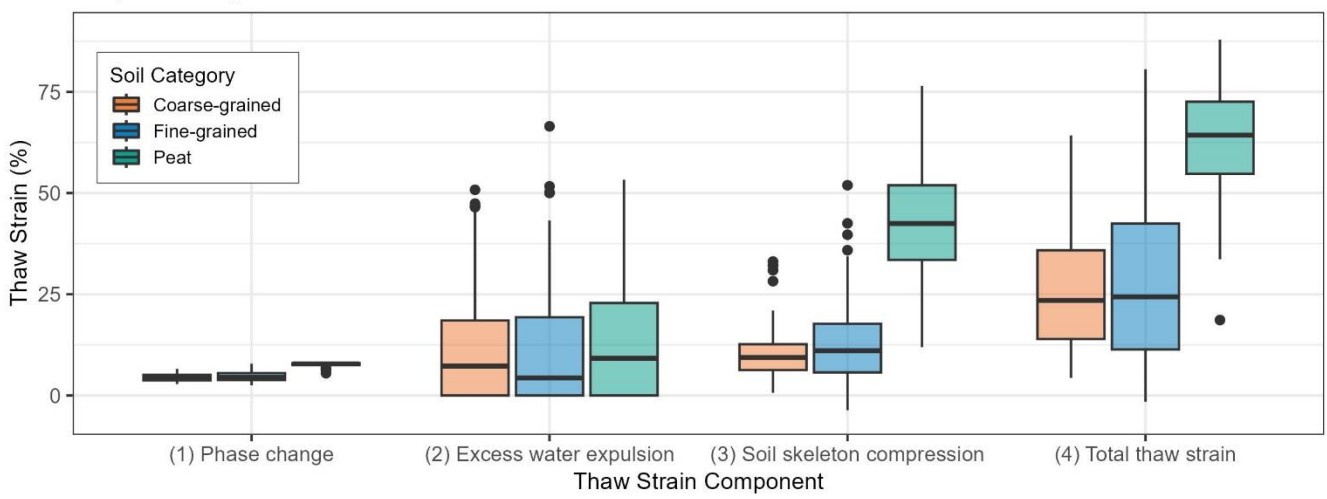

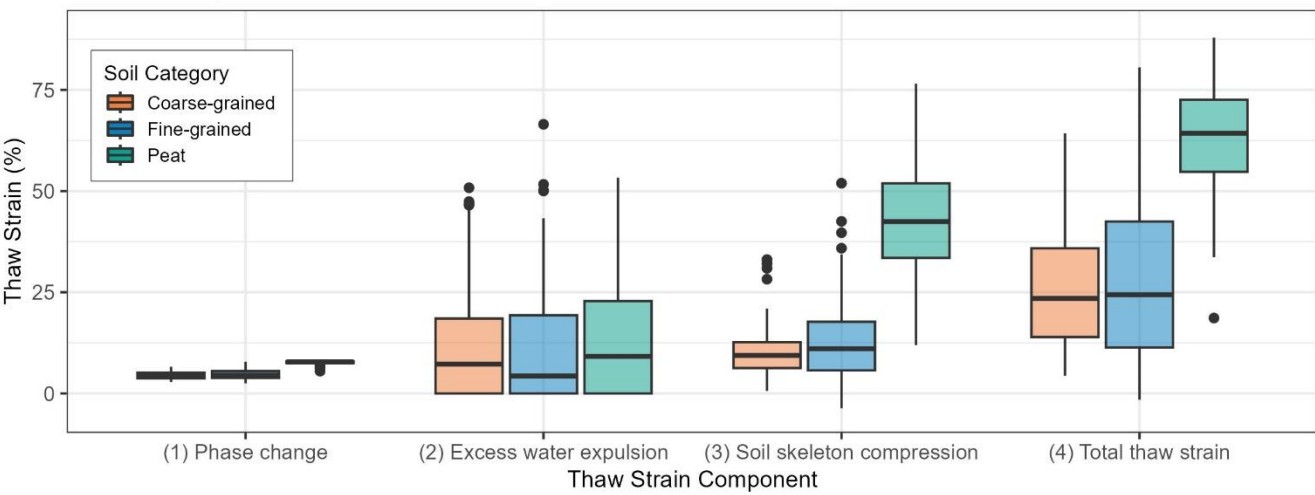

**Figure A. 1: Box plots showing the distribution of thaw strain components in ice-poor and ice-rich samples, with a comparison between three main soil groups in the PTS dataset**