# Peer review of "Compilation and Analysis of Thaw Settlement Test Results: Implications for Prediction Tools and Stress-Strain Characterization in Permafrost"

_Earth System Science Data, 2024_

## Author Comment (AC1)

Dear Reviewer,

We are grateful for the thoughtful and constructive feedback on our manuscript. Below, we provide point-by-point responses to each of the comments and describe how they have been addressed in the revised manuscript.

The provided comments helped us improve the clarity, framing, and focus of the manuscript. We believe the revised version better communicates the intended use, contributions, and limitations of the dataset and associated analyses.

To help distinguish between the comments and our responses, *the comments are shown in **black***, and our responses are shown in **Orange.**

***General comment:***

*"The paper presented a repository of thaw settlement test results sourced from the literature (generally sourced from Canada). The paper used the data in the literature to compare the effectiveness of existing empirical tools for estimating thaw strain and to identify the most fitting tools for various soil groups."*

***Comment 1:***
*"Although the tool presented in the paper will be useful for preliminary assessments, I'd like to point out that there is still a need for site specific data. If we don't understand the underlying foundation conditions for the transportation infrastructure (i.e. example given in the paper), the estimates resulting from any of these methods may be completely off. This was demonstrated in Figure 12 where thaw strains are generally higher than the curve estimates."*

**Response:**
We agree with the reviewer that site-specific conditions are essential for accurate thaw settlement predictions. This point is now more clearly stated in the Abstract (lines 29–30) and in Section 6: Uncertainty and Limitations (lines 610–615), where we emphasize that the dataset supports thaw strain estimation at the material level and must be complemented with site-specific data, including local stratigraphy and thaw depth, to estimate total settlement.

***Comment 2:***
*"While the paper did not address it, there is also a possibility that water can infiltrate through thawing layers and exacerbate 'still frozen' layers, which in turn can affect the development of thaw settlements."*

**Response:**

We thank the reviewer for this insightful observation. A note has been added in Section 6 (lines 614–619) to acknowledge that coupled hydrological and thermal processes, such as water infiltration accelerating deeper thaw, may influence ground response. These processes are not captured by the dataset, which focuses on strain-based responses at the material scale.

*Comment 3:*

*"The paper can be improved by removing some redundancy on the 'benefits,' which was repeated several times throughout the paper, and placing a strong statement either at the beginning or at the end of the paper."*

**Response:**

We appreciate this suggestion and have revised the manuscript to reduce repetition, particularly in the Conclusions section (line 640 onward), where the key contributions of the dataset are now summarized more concisely. Additionally, the Abstract and Section 6 have been revised to more clearly define the scope and limitations of the dataset, helping avoid overstatement of its applications.

*Comment 4:*

*"The authors have used the context of climate change as a possible precursor for future issues with thaw settlements. It would be beneficial to the reader, and to the improvement of the paper, if the authors can highlight or provide an example how the data they have available in the PTSD can be used to 'predict' future settlements. For any of the data points available, assuming it was frozen at the time it was sampled at a certain depth, what would be the expected settlement today (if a temperature threshold is breached, and considering the recorded climate in the last 30 years)?"*

**Response:**

We thank the reviewer for raising this important point. In Section 6 (from line 609), we now clearly define the intended scope of the dataset. While the data enable thaw strain estimation based on material properties, total thaw settlement also depends on the thickness of the thawed layer, which varies across sites and over time. We now explicitly state that the dataset is not intended to predict future settlement on its own but can support such analyses when used in combination with site-specific or modeled thaw depth inputs.

***Comment 5:***

*"Minor typographical errors: Figure 1.a should be 2.a in Line 104; Line 179, Sec. 0?"*

**Response:**

Thank you for pointing these out. Both issues have been corrected in the revised manuscript.

---

## Author Comment (AC2)

Dear Reviewer,

We are grateful for the detailed and thorough review. The comments provided helped identify areas for improvement in both the dataset and the manuscript. These revisions have strengthened the overall quality of the work. We greatly appreciate the time and effort put into the review.

Below is a point-by-point response explaining how we addressed each comment in the revised manuscript and dataset.

To help distinguish between the comments and our responses, *the comments are shown in **black**,* and our responses are shown in **Orange**.

**Point-by-point response:**

*"This article aggregates and standardizes thaw settlement test data from a variety of sources, some of which previously unavailable, to allow for meaningful comparison. Additionally, the authors demonstrate the utility of this new dataset by benchmarking empirical estimates of thaw strain behaviour."*

*"Slightly more discussion of the quality control process would benefit a user of this data. More specifically, for data obtained using PlotDigitizer, is there an estimate on accuracy and precision? "*

Response:  An explanation was added clarifying the expected accuracy of digitized data in Section 3 (line 190-193) and also Section 6 (line 591-592).

*"Similarly, in section 3, it would be valuable to elaborate on the quality control steps and the implications of the quality levels for someone who intends to use the data. For instance, under what circumstances (if any) could level 0 or 1 data be used? "*

*Response:  An explanation was added to Section 3 (line 220-224) and Section 6 (line 586-588) clarifying the intended use of each data quality level.*

*"While it is mentioned in the paper, additional emphasis could be placed on the calculation of A0, a0, Cc*, e*th as explicit solutions to the problem of non-standardized tests. Making it*

*clear to the reader early on that "These parameters form [the] basis for future comparisons of thaw settlement characteristics across various soil types" would serve to highlight one of the major contributions of this work."*

Response:  A sentence was added in Section 2 (line 100-102) to emphasize that these derived parameters address the lack of standardization and form the basis for cross-comparison of thaw settlement behavior.

*"Finally, a number of specific technical changes will make the data easier to use, and improve the clarity of the manuscript:*

***Data and repository***

*------------------*
*\* In the git repository, consider mentioning which version of R was used for greater repeatability."*

Response:  The version of R used has been added to the GitHub repository README.

**-borehole_location_and_datasource**

*"\* do latitude and longitude values have an associated datum?  Is there one assumed?  Mention this in README metadata and/or in article*

*\* What is the estimated precision of lat/lon coordinates? Comment on this in S3.4 or S6. Some from 1976 are listed with six decimal places. Have these been re-surveyed? "*

Response:  An explanation of the coordinate format used in the original reports has been added to the article (Section 3.4 and Section 6) and dataset README. Decimal places have been adjusted to better reflect the actual precision of the recorded coordinates.

*"\* Values of ref column do not exactly match fields in the 'Citation' column of data_sources. This makes it difficult to automatically connect access details to boreholes to samples. "*

Response:  The column containing full citations has been renamed to data_source across all files. Citation values have been standardized to ensure consistency throughout the dataset.

*-thaw_settlement_test_result*

*"* The 'unique_id' column might be better described as a 'sample_id' (also in sample_details and particle_size). Not only it is more descriptive, but 'unique' id in this context gives the impression that each row would have a different value which is not the case in this table or in particle_size"*

Response:  sample_name is not unique in all cases, as multiple tests were sometimes performed on subsamples from the same core or block. Therefore, unique_id and sample_name are not one-to-one. Since each entry represents a single thaw settlement test, unique_id was renamed to test_id to make the naming more descriptive and consistent.

"* I assume 'deformation' is 'relative to frozen sample?'  if so, add to README metadata description"

Response: The recorded deformation represents cumulative vertical deformation (in mm) measured at the end of each load step, relative to the initial (frozen) state of the sample. This description has been clarified in the dataset README file.

*-sample_details_with_derived_parameters*

*"* it appears at first glance that unique_id and sample_name form a one-to-one correspondance. If this is the case, and if you think it simplifies things, consider using sample_name as your unique ID (keeping the sample_name column). Otherwise, clarify whether unique_id and sample_name are indeed one-to-one in article"*

Response: Each entry in the dataset represents a single test result. While some sources provided explicit test-level identifiers, others used only sample-level naming without assigning specific test IDs. Since multiple tests may be conducted on subsamples from the same core or block, we constructed test names in these cases by appending suffixes (e.g., _1, _2) to the sample name. As a result, test_name is unique. However, because some generated names were lengthy, we introduced a short, unique test_id (previously unique_id) to serve as the primary identifier across all files.

*"* sample_volume_init is reportedly in mm3. However, values of ~800 mm3 with heights of ~100mm imply very narrow cross-sections. Is this correct?"*

Response: The correct unit (cm$^3$) has been added in the dataset README file.

*"* sample_diameter_ave_or_typ :  units?"*

Response: The unit (mm) has been added to the dataset README file.

*"* note: typo in "sewelling""*

Response: The typo has been corrected.

***-data_sources_and_access_details***

*"* Are these reports accessible online anywhere other than the google drive?  Google drive links can be somewhat ephemeral.  One option could be to store the reports also in a more long-term repository (zenodo or other archive). this would depend on original authors."*

Response: Thank you for the suggestion. Google Drive links have been replaced with stable DOIs.

*"* Consider omitting 'row' column or renaming to something more descriptive (source_id ?)"*

Response: The column was renamed to source_id, which contains a short identifier for each original data source and is used consistently across all dataset files.

*"* column names are not in same style as other csv files"*

Response: Column names have been revised to align with the naming style used in the other CSV files in the dataset.

*"* README file metadata incomplete for this csv file"*

Response: Descriptions of key columns in the data_sources_and_access_details.csv file have been added to the dataset README.

**Typos and Style suggestions**

\-\-\-\-\-\-\-\-\-\-\-\-\-\-\-\-\-\-\-\-\-\-\-\-\-

*"L11: "Infrastructures" in plural. Is this a typo?  Elsewhere in the paper it is always singular.*

*L68: Lewkowicz et al 2024 definition for thaw settlement: this glossary and definition is intended as a plain-language summary. Is there a more technical source?*

*L73: "obtained through this test" could be omitted for clarity*

*L80: in caption (and in other captions) references lack bold typeface*

*L82: "The thaw settlement [test] typically"*

*L90-95: I believe the notation (e - \sigma_v) is meant to represent the two axes of the plots described. However, it is easy to read this as subtraction when reading.  Is this notation common in engineering?  If not, consider explicitly introducing: e.g. "… presented as either thaw strain versus vertical (effective) stress (hereafter referred to as a $\varepsilon - \sigma v$ curve) … ", or using clearer notation.*

*L136: missing period.*

*L179: "Sec. 0"*

*L194: "indicated by ODS in Figure 4" or omit parenthetical material*

*L287: consistency in units: negative exponent vs fraction for m3. Also period after kg*

*L325: can you be more specific about which of "the methods outlined in Sec. 2.2" you use?*

*L387: capitalize Appendix*

*L585: Consider adding a memorable named tag (e.g. 'essd-pts', 'to commit 5678ccd  (https://git-scm.com/book/en/v2/Git-Basics-Tagging)"*

\-\-\-\-\-\-\-\-\-\-\-\-\-\-\-\-\-\-\-\-\-\-\-\-\-

Response:  The following edits were made as suggested:

*L11, L73, L80, L82, L90–95, L136, L179, L194, L287, L325, L387

*A more technical definition of thaw settlement was added to the first paragraph of Section 2. (Previously L68)

*A more descriptive GitHub tag (essd-pts-final) was used for the commit and mentioned in the code availability statement.